# Proteomics of immune cells from liver tumors reveals immunotherapy targets

## Graphical abstract

## Authors

Fernando P. Canale, Julia Neumann, Janusz von Renesse, ..., Nuh N. Rahbari, Giorgio Ercolani, Roger Geiger

## Correspondence

roger.geiger@irb.usi.ch

## In brief

Canale et al. employed mass spectrometry-based proteomics to analyze immune cells from blood, liver, and tumor tissues of 48 hepatocellular carcinoma patients. They identified significant phenotype alterations in tumor-infiltrating immune cells, including SGPL1 upregulation in macrophages and AFAP1L2 in T cells. Genetic deletion of these genes in murine macrophages and T cells, respectively, enhanced anti-tumor activity, indicating their potential as immunotherapy targets.

## Highlights

- Proteomes of FACS-purified T cells, NK cells, and macrophages from 48 HCC patients

- Tumor-infiltrating CD8 T cells upregulate AFAP1L2 in response to chronic stimulation

- Deletion of AFAP1L2 in CD8 T cells enhances their anti-tumor activity

- Tumor macrophages upregulate SGPL1, which inhibits inflammatory anti-tumor functions

 Canale et al., 2023, Cell Genomics 3, 100331
June 14, 2023 © 2023 The Author(s).

CellPress

# Proteomics of immune cells from liver tumors reveals immunotherapy targets

Fernando P. Canale,[1,2,17] Julia Neumann,[1,2,17] Janusz von Renesse,[3,4] Elisabetta Loggi,[5] Matteo Pecoraro,[1,2] Ian Vogel,[1,2] Giada Zoppi,[1,2] Gaia Antonini,[1,2] Tobias Wolf,[1,2] Wenjie Jin,[1,2] Xiaoqin Zheng,[1,2] Giuliano La Barba,[6] Emrullah Birgin,[7] Marianne Forkel,[8] Tobias Nilsson,[8] Romina Marone,[9,10] Henrik Mueller,[8] Nadege Pelletier,[8] Lukas T. Jeker,[9,10] Gianluca Civenni,[2,11] Christoph Schlapbach,[12] Carlo V. Catapano,[2,11] Lena Seifert,[3,4] Adrian M. Seifert,[3,4] Silke Gillessen,[2,13] Sara De Dosso,[2,13] Alessandra Cristaudi,[14] Nuh N. Rahbari,[7] Giorgio Ercolani,[15,16] and Roger Geiger[1,2,11,18,*]

[1]Institute for Research in Biomedicine (IRB), Bellinzona, Switzerland
[2]Università della Svizzera italiana (USI), Lugano, Switzerland
[3]Department of Visceral, Thoracic and Vascular Surgery, University Hospital Carl Gustav Carus, Technische Universität Dresden, Dresden, Germany
[4]National Center for Tumor Diseases (NCT), Dresden, Germany
[5]Hepatology Unit, Department of Medical & Surgical Sciences, University of Bologna, Bologna, Italy
[6]General and Oncologic Surgery, Morgagni-Pierantoni Hospital, Forlì, Italy
[7]Department of Surgery, Medical Faculty Mannheim, Universitätsmedizin Mannheim, Heidelberg University, Mannheim, Germany
[8]Roche Pharma Research and Early Development, Infectious Diseases Discovery, Roche Innovation Center Basel, Basel, Switzerland
[9]Department of Biomedicine, University of Basel, Basel, Switzerland
[10]Transplantation Immunology and Nephrology, Basel University Hospital, Basel, Switzerland
[11]Institute of Oncology Research (IOR), Bellinzona, Switzerland
[12]Department of Dermatology, Inselspital, Bern University Hospital, University of Bern, Bern, Switzerland
[13]Oncology Institute of Southern Switzerland (IOSI), Ente Ospedaliero Cantonale (EOC), Bellinzona, Switzerland
[14]Department of General and Visceral Surgery, Cantonal Hospital Lugano, Lugano, Switzerland
[15]Department of Medical and Surgical Sciences - DIMEC; Alma Mater Studiorum - Univeristy of Bologna, Bologna, Italy
[16]Morgagni-Pierantoni Hospital, Ausl Romagna, Forlì, Italy
[17]These authors contributed equally
[18]Lead contact
*Correspondence: roger.geiger@irb.usi.ch

## SUMMARY

Elucidating the mechanisms by which immune cells become dysfunctional in tumors is critical to developing next-generation immunotherapies. We profiled proteomes of cancer tissue as well as monocyte/macrophages, CD4$^+$ and CD8$^+$ T cells, and NK cells isolated from tumors, liver, and blood of 48 patients with hepatocellular carcinoma. We found that tumor macrophages induce the sphingosine-1-phospate-degrading enzyme SGPL1, which dampened their inflammatory phenotype and anti-tumor function *in vivo*. We further discovered that the signaling scaffold protein AFAP1L2, typically only found in activated NK cells, is also upregulated in chronically stimulated CD8$^+$ T cells in tumors. Ablation of *AFAP1L2* in CD8$^+$ T cells increased their viability upon repeated stimulation and enhanced their anti-tumor activity synergistically with PD-L1 blockade in mouse models. Our data reveal new targets for immunotherapy and provide a resource on immune cell proteomes in liver cancer.

## INTRODUCTION

Tumors evolve as dynamic ecosystems consisting of cancer, stromal, and immune cells. The immune infiltrate typically consists of various cell types including cytotoxic T cells that play a central role in the immune response to tumors. Stimuli in the tumor microenvironment (TME) can suppress the functionality of T cells, which then lose the ability to control tumor growth.[2–4] To restore T cell functionality in tumors, checkpoint inhibitors (CPIs) have been developed that block inhibitory receptors on T cells.[5,6] CPIs have yielded great successes in clinical settings,

but they are not always effective. This raises the need to identify novel strategies to further enhance immune responses to tumors.

One potential avenue to therapeutically improve anti-tumor immunity is to identify and target new molecular regulators underlying T cell dysfunction in tumors. Another possibility is to target tumor-associated macrophages (TAMs), which influence T cell function, i.e., through the production of cytokines that affect T cell differentiation. In many tumor types, TAMs have a tumor-promoting phenotype and are associated with poor prognosis.[7,8] Therefore, therapeutic strategies include depletion of

TAMs or functional reprogramming of tumor-promoting TAMs into inflammatory macrophages that mediate immune responses to tumors. To devise therapeutic interventions that enhance T cell functionality or reprogram macrophages, it is important to understand how immune cells are regulated in tumors.

Proteins are directly involved in most biological processes; hence, their quantification is important to understand the phenotype of a cell. As there is no simple relationship between transcripts and proteins, their abundances do not always correlate. For example, protein abundance is influenced by the translation efficiency of a given mRNA, by the stability of the protein, as well as by posttranslational modifications affecting localization and degradation.[9] Mass spectrometry (MS)-based proteomics enables the direct quantification of thousands of proteins within cells or tissues.[9,10] Owing to advances in all areas of the MS-based workflow, including sample preparation, chromatography, MS instrumentation, and data analysis, it is possible to determine deep proteomes of clinical samples for target identification.[11] Previous studies employed transcriptomics or antibody-based protein quantification to profile tumor-infiltrating immune cells,[12–15] yet mass spectrometry-based proteomic profiling of immune cells from a large cancer patient cohort is lacking. Hence, a detailed proteomic analysis of immune cells in tumors offers important new insights to identify putative immunomodulatory interventions to treat cancer.

In this study, we profiled the proteomes of cancer and immune cells from patients with hepatocellular carcinoma (HCC), for which there are currently few treatment options.[16] Although HCC is typically infiltrated by immune cells, CPIs are only partially effective.[17,18] HCC can be caused by chronic infection with hepatitis B (HBV) or C (HCV) viruses or different metabolic and inflammatory disorders related to non-alcoholic and alcohol-related steatohepatitis.[19–21] Our proteomic data revealed that TAMs upregulate SGPL1, which we found inhibited their anti-tumor activity. We also identified AFAP1L2 as a new target for T cell-based immunotherapies. AFAP1L2 was expressed specifically in chronically stimulated CD8+ T cells in tumors, and its genetic ablation improved anti-tumor activity.

## RESULTS

### Proteomic profiling of an HCC patient cohort

To analyze proteomes of immune and cancer cells from liver tumors, we prospectively obtained tumor tissue, adjacent liver tissue, and peripheral blood from patients with HCC undergoing surgical resection. Of 48 patients, 9 had a previous infection with HBV, 20 with HCV, and 19 had no history of viral liver infection (Table S1). From each patient, blood, dissociated liver, and tumor tissues were subjected to Ficoll gradient centrifugation to separate mononuclear immune cells from which memory CD4+ T cells, memory CD8+ T cells, CD56+ NK cells, and CD14+ monocytes/macrophages were sorted to a purity of >98% (Figures 1A and S1). The numbers of immune cells isolated from tumors varied considerably between patients and ranged from <100,000 to 6 million cells per gram of tumor tissue (Figure 1B).

From 32 tumor and 33 liver cell suspensions, we recovered a cell pellet from the Ficoll gradient, which contains hepatocytes and malignant and stromal cells (Figure 1A). After homogenization and lysis of cell pellets with 4% SDS, proteins were precipitated, digested, and analyzed by liquid chromatography coupled mass spectrometry (LC-MS). A total of 8,182 protein groups were quantified with an average of 5,209 protein groups per sample (Figure S2A, Table S2). A differential abundance analysis between liver and tumor tissues revealed profound alterations in tumor proteomes (Figures 2A and 2B). Data are accessible through our interactive platform www.immunomics.ch/hcc; an example is shown in Figure S2B. Tumor samples had typical molecular features of HCC based on a comparison to tumors from a different HCC patient cohort[22] (Figure 2A). In our data, PYCR2 was the most strongly upregulated protein in HCC. This enzyme catalyzes the formation of hydroxyproline, a major component of collagen that is associated with HCC tumor progression[23] (Figure 2C). Two additional proteins that mediate metabolic flux toward hydroxyproline, P4HA1 and P4HA2, were also strongly upregulated in HCC. Among the proteins that were most downregulated in HCC was ASS1, which is involved in arginine biosynthesis (Figure 2C), consistent with previous findings.[24] In summary, our proteomic analysis of immune cell-depleted liver and tumor tissues confirmed correct classification of tumorous and non-tumorous samples and recapitulated characteristic proteomic alterations in HCC tumors,[22] highlighting metabolic alterations related to hydroxyproline.

Next, we processed a total of 497 immune cell samples for proteomic analysis according to previously established protocols.[25] A total of 9,790 protein groups were quantified with an average of 5,832 protein groups per sample (Figure 2D). All data are accessible through www.immunomics.ch/hcc. As expected, the lineage markers CD4, CD8, and CD56 were most abundant in the respective immune cell types, and the macrophage marker MRC1 was highest in liver and tumor macrophages (Figure 2E). Consistent with previous findings,[26,27] the exhaustion marker CD39 was upregulated in tumor-infiltrating CD4+ and CD8+ T cells and macrophages compared with cells isolated from adjacent liver tissue and blood (Figure S2C). Thus, the proteome data were in agreement with our cell sorting strategy and captured known regulations in tumor immunology.

### Tumor macrophages display THY-1, acquired from the environment

To analyze phenotypic changes in macrophages that infiltrate tumors, we performed a differential abundance analysis between proteomes of liver macrophages and TAMs and identified 15 proteins that were strongly increased in TAMs (Log2 fold change > 2; p value < 0.0001) (Figures 3A and 3B). In addition, we analyzed transcriptomes of tumor macrophages from three HCC patients by RNA-seq and estimated copy numbers of transcripts, as previously described.[28] A comparison of transcript and protein copy numbers revealed a median protein-per-mRNA ratio of 500:1, but this ratio varied considerably (Figure 3C). For example, the chemokines CXCL8 and CCL3, which are secreted by macrophages, had a ratio lower than 10:1. In contrast, the chemokine CCL15 had a protein-per-transcript ratio greater than 10,000:1, because it is not expressed by macrophages but is taken up from the environment. Macrophages express the CCR1 receptor, which captures CCL15 secreted by liver cells.[29]

**CellPress**

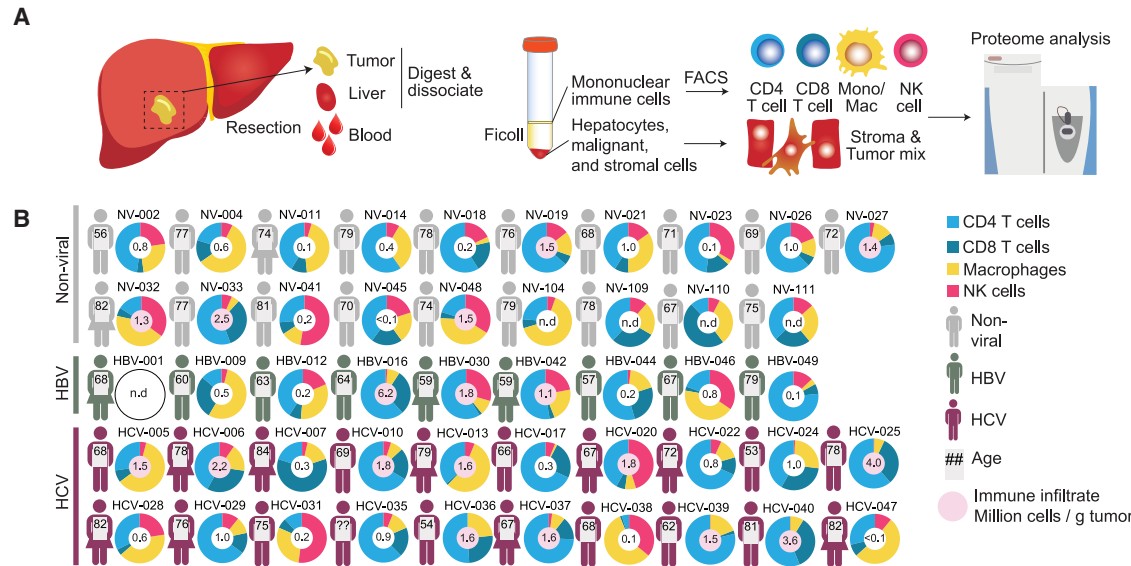

**Figure 1. Schematic of workflow and patient cohort**
(A) Schematic overview of this study.
(B) Overview of the patient cohort. Age and sex are indicated. Pie charts show the percentages of CD4[+] T cells, CD8[+] T cells, NK cells, and macrophages that were isolated from HCC tumors. In the center of the pie chart the total number of T cells, NK cells, and macrophages that were isolated from tumors is shown. Cases in which more than a million immune cells/g tumor were isolated are marked in red. See also Figure S1 and Table S1.

Among the most enriched proteins in TAMs, THY-1, AKR1B10, and GGH had a protein-per-mRNA ratio greater than 10,000:1, suggesting they were acquired from the TME. AKR1B10, a potential diagnostic marker of HCC,[30] and GGH are both soluble metabolic enzymes that were likely taken up by macrophages through macropinocytosis. In contrast, THY-1 (CD90) is a GPI-anchored membrane protein that is abundant on the surface of cancer stem cells, fibroblasts, and endothelial cells[31] but has not been found to be expressed by macrophages. We confirmed the presence of THY-1 protein on tumor macrophages by flow cytometry (Figure S3A) and the absence of *THY-1* mRNA by RNA *in situ* hybridization (Figure S3B).

To confirm THY-1 protein transfer, we incubated blood-derived monocytes with patient-matched HCC tumor tissue fragments for 6 days and found that monocytes indeed acquired THY-1 protein from the TME (Figures S3C and S3E). Collectively, these data demonstrate that macrophages in liver tumors acquire proteins; hence, their phenotype is not exclusively shaped by transcription but also through interactions with the environment.

### Deletion of SGPL1 promotes inflammatory macrophages that inhibit tumor growth

Some of the 15 proteins that are enriched in tumor macrophages may render them anti-inflammatory and contribute to a tumor-promoting phenotype. We therefore assessed whether any of these 15 proteins are upregulated in functionally polarized M2 macrophages, which represent a model for anti-inflammatory macrophages. For this, freshly isolated monocytes from four healthy donors were *in vitro* polarized to an M1 (LPS + INF-γ; inflammatory) and an M2 phenotype (IL-4 and IL-13; anti-inflammatory), and their proteome was analyzed by LC-MS (Table S3).

A differential abundance analysis confirmed strong enrichment of typical markers, such as GBP4 and GBP5 in M1 macrophages and CD209 and ALOX15 in M2 macrophages (Figures 3D and S3F). Of the 15 proteins upregulated in tumor macrophages, DAB2, SGPL1, and GLUL were enriched in M2 macrophages. DAB2 and GLUL were previously found to be upregulated in TAMs and M2 macrophages, and genetic ablation of *Dab2* or *Glul* in murine macrophages was found to improve their anti-metastatic function.[32,33] However, SGPL1 (sphingosine-1-phosphate [S1P] lyase) has previously not been associated with TAMs, and its function in macrophages is unknown.

SGPL1 is an endoplasmic reticulum membrane protein that irreversibly degrades S1P, a sphingolipid involved in immune signaling and inflammation.[34] To study the function of SGPL1 *in vivo*, we generated bone marrow-derived macrophages (BMDMs) from Rosa-Cas9 mice and lentivirally transduced them with two different sgRNAs targeting *Sgpl1* (Figure 3E). Tracking of indels by decomposition (TIDE) analysis[35] confirmed that the *Sgpl1* locus was edited in 94% and 87% of BMDMs, respectively (Figure S3G). We first assessed *in vitro* the capacity of *Sgpl1*-edited BMDMs to upregulate the costimulatory receptor CD86 and to produce inflammatory cytokines. For this, we stimulated BMDMs with LPS and IFN-γ and then used flow cytometry to analyze the abundance of intracellular IL-6 and IL-12, as well as surface CD86. *Sgpl1*-edited BMDMs displayed more CD86 on their surface and more frequently produced IL-6 and IL-12 than control BMDMs (Figures 3F–3H). An analysis of cell supernatants by ELISA showed that *Sgpl1*-edited BMDMs secreted 7–11 times more IL-12 than control BMDMs (Figure 3I). This confirms that knockout of *Sgpl1* increases IL-12 production, which drives anti-tumor immunity.[36]

**Figure 2. Characteristics of HCC and immune infiltrates**

(A) Volcano plot from differential abundance analysis (two-tailed Welch's t test) between proteomes of non-tumorous (n = 33) and tumorous liver tissue (n = 32). Each dot represents a protein. Signature proteins of HCC tumors identified in a previous proteomics study[22] are indicated in dark blue. See also Figure S2A and Table S2.

(B) Heatmap showing the abundance (Z score) of the five most strongly up- and downregulated proteins in HCC tissue in individual patients.

(C) Differential analysis of the arginine and proline metabolism between non-tumorous and tumorous tissue. Enzymes are color-coded according to the fold change as determined by the differential abundance analysis in (A).

We then asked whether *Sgpl1*-edited BMDMs promote anti-tumor immunity *in vivo*. To test this, we mixed MC38 tumor cells together with *Sgpl1*-edited BMDMs in a 1:1 ratio and co-injected them into wild-type recipient mice (Figure 3J). As a benchmark, we performed the same experiments with *Dab2*-edited BMDMs (editing efficiency of 67%, Figure S3H), which are known to have an anti-tumor effect. After 1 week, *Sgpl1*-edited BMDMs significantly inhibited tumor growth compared with control BMDMs (Figure 3J). The extent of tumor growth inhibition was comparable to *Dab2*-edited BMDMs. Taken together, our data indicate that macrophages upregulate SGPL1 in tumors, which in a mouse model reduced their inflammatory response, and impaired their anti-tumor activity.

### Tumor NK cells upregulate AFAP1L2

A differential abundance analysis between proteomes of NK cells isolated from liver and tumor tissues identified four proteins that were increased in tumor NK cells (Log2 fold change > 2; p value < 0.01): AKR1B10, GLUL, IGHMBP2, and AFAP1L2 (Figures 4A and 4B). A further comparison with blood NK cells revealed that only the potential HCC marker protein AKR1B10, which was also enriched in TAMs (Figures 3A and 3B), and AFAP1L2, a cytosolic signaling scaffold protein,[37] were specifically upregulated in tumor-infiltrating NK cells. Interestingly, inspection of a previously published human immune cell proteome atlas[38] revealed that AFAP1L2 was exclusively expressed in activated NK cells, while it was not present in resting NK cells or any other immune cell type at steady state or upon canonical activation (Figure S4A). Taken together, AFAP1L2 is upregulated in activated, tumor-infiltrating NK cells.

### AFAP1L2 is induced in TILs likely due to chronic stimulation

We next analyzed T cells by flow cytometry and found that only one-third of HCC patients displayed a high frequency of PD1+ CD8+ T cells in tumors (Figure 4C). 14 out of 15 of these patients had a viral etiology of disease (HBV or HCV), while 22 out of 36 of the PD-1neg/low patients had a non-viral etiology of disease. We then compared proteomes of CD8+ tumor-infiltrating lymphocytes (TILs) of which greater than 60% were PD-1+ (10 patients) to proteomes of CD8+ TILs of which less than 20% were PD-1+ (15 patients). This differential abundance analysis revealed 22 proteins that were strongly upregulated in CD8+ TILs with a high frequency of PD-1+ T cells (log₂ fold change > 3, p value < 0.01) (Figures 4D and 4E). This signature that correlated with high PD-1 expression included the nuclear factor TOX, which promotes T cell exhaustion,[39–41] the surface protein CD38, which is a marker for dysfunctional T cells,[42] as well as UBASH3B (STS2), a negative regulator of TCR signaling.[43] Interestingly, several proteins associated with proliferation were upregulated, such as seven DNA replication licensing factors (MCM2-7), SMC2, a central component of the condensin complex, and the proliferation marker Ki-67 (Figures 4D and 4E).

Consistent with the notion that PD-1 marks proliferating T cells, we recovered significantly more CD8+ T cells from tumors when the percentage of CD8+ T cells expressing PD-1 was high (Figure 4F). These data suggest that PD-1+ CD8+ T cells in HCC tumors can be highly proliferative, consistent with previous observations in melanoma tumors.[13]

We asked which of the 22 PD-1-associated signature proteins were induced by T cell activation. For this, we isolated naive CCR7+CD45RA+ CD8+ T cells from the blood of four healthy donors, activated them with plate-bound antibodies to CD3 and CD28 for 48 h, and then cultured them for an additional 48 h. Samples were collected after increasing times and analyzed by LC-MS (Table S4). We found that the abundance of 19 out of the 22 "PD-1 signature proteins" was strongly increased after T cell activation (Figure 5A). However, TOX, LMCD1, and AFAP1L2 were either not detected or only at very low levels, suggesting that their presence in tumor-infiltrating T cells was not a consequence of the canonical T cell activation program.

Because tumor-specific CD8+ TILs are chronically stimulated at the tumor site, we hypothesized that TOX, LMCD1, and AFAP1L2 might be induced upon chronic stimulation. To test this, we continuously stimulated naive CD8+ T cells for 14 days with plate-bound antibodies to CD3 and CD28. For comparisons, we activated naive CD8+ T cells for only 3 days and then either cultured them for 11 days or re-stimulated them after 8 days (Figure 5B). Markers associated with T cell exhaustion such as CD39, PD-1, TIM-3, and LAG3 were consistently increased in chronically stimulated T cells when analyzed by flow cytometry (Figures 5B and S4B), indicating that upon extensive *in vitro* stimulation, CD8+ T cells acquired phenotypic properties similar to TILs.

We then analyzed transiently activated, re-activated, and chronically stimulated T cells by proteomics and found that the 19 "PD-1 signature proteins" that were upregulated by T cell activation were maintained at similar abundances following chronic stimulation (Figure 5C, Table S5). Out of the three proteins (TOX, LMCD1, and AFAP1L2) that were not induced upon transient activation, AFAP1L2 was strongly induced upon chronic stimulation (Figure 5C). We estimated protein copy numbers in CD8+ T cells and found that transiently activated T cells contained close to zero AFAP1L2 protein copies (Figure 5D). However, re-activated T cells contained ~100,000 copies of AFAP1L2 protein, and chronically stimulated cells had ~300,000 copies (Figure 5E). Consistent with this, we observed only background levels of AFAP1L2 mRNA in transiently activated CD8+ T cells, but the abundance of AFAP1L2 mRNAs was considerably higher in re-activated T cells and further increased upon chronic stimulation (Figure 5F, Table S6).

Since AFAP1L2 is only expressed in T cells upon repeated stimulation, it could serve as a specific target to exclusively reinvigorate chronically stimulated T cells at the tumor site. We analyzed publicly available single-cell RNA-seq datasets of CD4+ and CD8+ T cells isolated from tumors of patients with HCC,[12] colorectal cancer,[44] and non-small cell lung cancer.[45]

---

(D) FACS-purified CD4+ and CD8+ T cells, CD14+ monocytes/macrophages, as well as NK cells from blood, liver, and tumor tissue were analyzed by LC-MS. The plots show the number of identified proteins in each sample.

(E) Boxplots showing the abundance of CD4, CD8, CD56, and MRC1 protein in different cell types isolated from blood, liver, and tumor tissue. Each dot represents a sample from a different patient. See also Figure S2C.

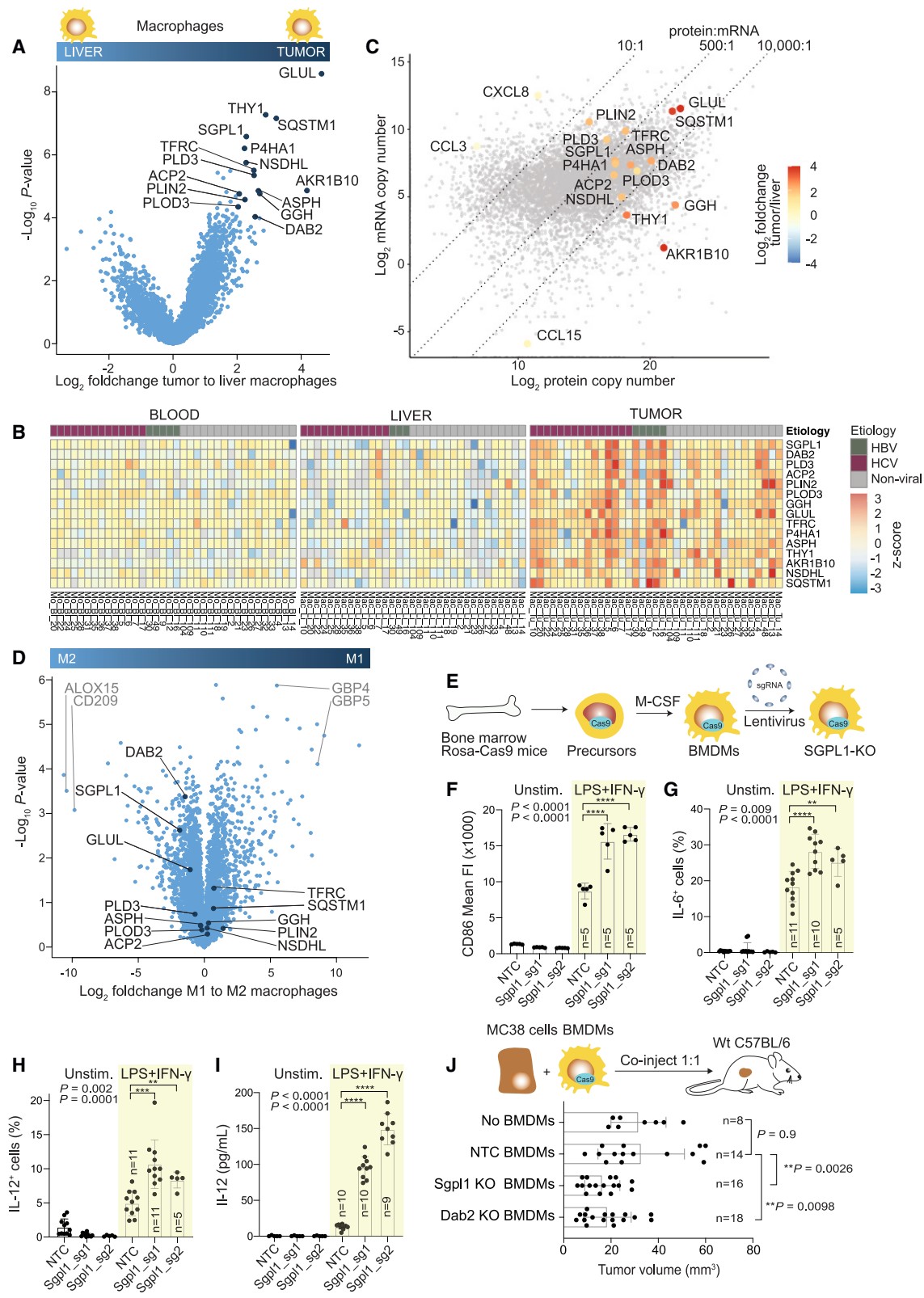

In all cancer types, AFAP1L2 was present in clusters of PD-1[+] CD8[+] T cells that express exhaustion markers such as LAG3, TIM-3, and CD39 (Figure S4C). Notably, these exhaustion markers were also found in CD4[+] Tregs, whereas AFAP1L2 was exclusively found in exhausted CD8[+] T cells. Collectively, these data indicate that AFAP1L2 is induced in repeatedly stimulated CD8[+] T cells across different cancer types. Since the function of AFAP1L2 in T cells was unknown, we next characterized its functional impact.

### AFAP1L2 impairs T cell viability upon chronic stimulation

Upon chronic stimulation by tumor antigens, T cells undergo apoptosis, which plays a critical role in establishing tumoral immune resistance.[46] Since AFAP1L2 is a cytosolic signaling scaffold protein involved in the regulation of cell proliferation and survival,[37] we asked whether ablation of *AFAP1L2* influences proliferation and viability of CD8[+] T cells. We ablated *AFAP1L2* by CRISPR-Cas9 in primary human CD8[+] T cells using two different sgRNAs (editing efficiency of 77% and 59%, respectively, Figure S5). To analyze proliferation, we labeled T cells with CellTrace Violet (CTV), stimulated them for 5 days with antibodies to CD3 and CD28, and analyzed cells by flow cytometry. We did not observe any differences in proliferation (Figure 5G). Next, we chronically activated T cells for 8 and 12 days and subsequently quantified the number of live CD8[+] T cells by flow cytometry. Strikingly, upon chronic stimulation, the number of live *AFAP1L2*-edited CD8[+] T cells doubled across both sgRNAs (Figure 5H). Thus, AFAP1L2 does not impact proliferation but promotes cell death of chronically stimulated T cells and may therefore have a negative effect on tumor control.

### Knockout of *Afap1l2* in murine T cells improves antitumor functions

To study whether AFAP1L2 plays a role in the T cell response to tumors, we used CD8[+] OT-I T cells, which have a transgenic TCR recognizing the ovalbumin (Ova)-derived SIINFEKL peptide presented on the MHC-I allele H-2Kb (Figure 6A). Freshly isolated OT-I T cells were activated with plate-bound antibodies to CD3 and CD28, and after 24 h, they were electroporated with a plasmid encoding Cas9, GFP, and a sgRNA targeting *Afap1l2*.[47]

After 24 h, GFP[+] OT-I T cells were sorted, and the gene editing efficiency was assessed by TIDE analysis. We used two different sgRNAs to disrupt the *Afap1l2* gene (editing efficiency: 58% and 42%, Figure S6A). Upon chronic stimulation, the number of live *Afap1l2*-edited OT-I T cells increased 2-fold compared to non-targeting controls, phenocopying our observations in primary human T cells (Figure S6B).

Next, *Afap1l2*-edited OT-I T cells were re-stimulated for 48 h and then transferred into C57BL/6 mice with established B16.OVA tumors, which are recognized by OT-I T cells. 5 days later, we analyzed tumor infiltrates by flow cytometry and found that *Afap1l2*-edited OT-I T cells were more abundant in tumors than control OT-I T cells (18% vs. 10% of total CD8[+] T cells in tumors) (Figures 6B and 6C), indicating that *Afap1l2*-edited T cells mounted a more robust response. An intracellular cytokine staining of tumor-infiltrating T cells showed that the percentage of *Afap1l2*-edited OT-I T cells producing TNFα and IFN-γ was twice as high as in controls (Figures 6D and 6E). Taken together, these data indicate that *Afap1l2* knockout increases both the quantity and potency of the anti-tumor T cell response.

When we followed B16.OVA tumor sizes over time, we found that *Afap1l2-edited* OT-I T cells mounted a superior anti-tumor response than control T cells. The effect was persistent across two different sgRNAs targeting *Afap1l2* (Figures 6F and 6G). We additionally treated mice with PD-L1 blocking antibodies and found that PD-L1 blockade combined with ablation of *Afap1l2* in T cells synergistically reduced tumor growth and significantly increased the survival of mice (Figures 6H and 6I).

To extend our analysis to a different tumor type, we used the MC38-OVA model. MC38-OVA tumors induce a strong endogenous T cell response and are completely eradicated upon transfer of OT-I T cells. Since this setting does not allow analyzing the impact of *Afap1l2* ablation in T cells on tumor growth, we used *Cd3e*[−/−] mice[48] that lack all endogenous T cells. MC38-OVA tumor cells were subcutaneously injected into *Cd3e*[−/−] mice, and after 5 days, either control or *Afap1l2*-edited OT-I T cells were adoptively transferred. Under these conditions, *Afap1l2*-edited OT-I T cells had a superior anti-tumor activity than control OT-I T cells (Figure 6J). In summary, we found that chronically stimulated CD8[+] T cells induce AFAP1L2, which when ablated improves their anti-tumor functions.

**Figure 3. TAMs upregulate SGPL1, which impedes their inflammatory anti-tumor functions**

(A) Volcano plot from differential abundance analysis (two-tailed Welch's t test) between proteomes of macrophages isolated from non-tumorous liver tissue (n = 33) and tumorous liver tissue (n = 38). Proteins that are most strongly upregulated in tumor macrophages are highlighted.

(B) Heatmap showing the abundance of proteins that are significantly upregulated in TAMs across all samples from blood monocytes, liver, and tumor macrophages.

(C) Estimates of protein copy numbers (mean of n = 38) are plotted against mRNA copy numbers of TAMs (mean of n = 3). The color code represents the fold change of selected proteins between liver macrophages and TAMs. The ratios on top of the graph indicate the protein-per-mRNA ratio.

(D) Volcano plot from differential abundance analysis (two-tailed Welch's t test) between proteomes of *in vitro* polarized M1 macrophages (LPS + IFN-γ) and M2 macrophages (IL-4 + IL-13) (n = 4). Proteins that are upregulated in TAMs are shown as black dots. THY-1 and AKR1B10 were not identified in *in vitro* polarized macrophages. See also Figure S3F.

(E) Schematic of the procedure to generate knockout BMBMs.

(F–I) BMDMs from Rosa-Cas9 mice were lentivirally transduced with a non-targeting control (NTC) sgRNA or with two different sgRNAs targeting *Sgpl1*. Then, BMDMs were left untreated or were stimulated with LPS + IFN-γ. After 16 h, CD86 (F), intracellular IL-6 (G), and intracellular IL-12 (H) were analyzed by flow cytometry and secreted IL-12 by ELISA (I).

(J) 1 week after co-injecting BMDMs and MC38 cells (500,000 cells, each), tumor sizes were measured. (F–J) P values and the number of samples are shown in graphs and were determined using a two-tailed t test. Bars represent means ± SEM; two independent experiments.

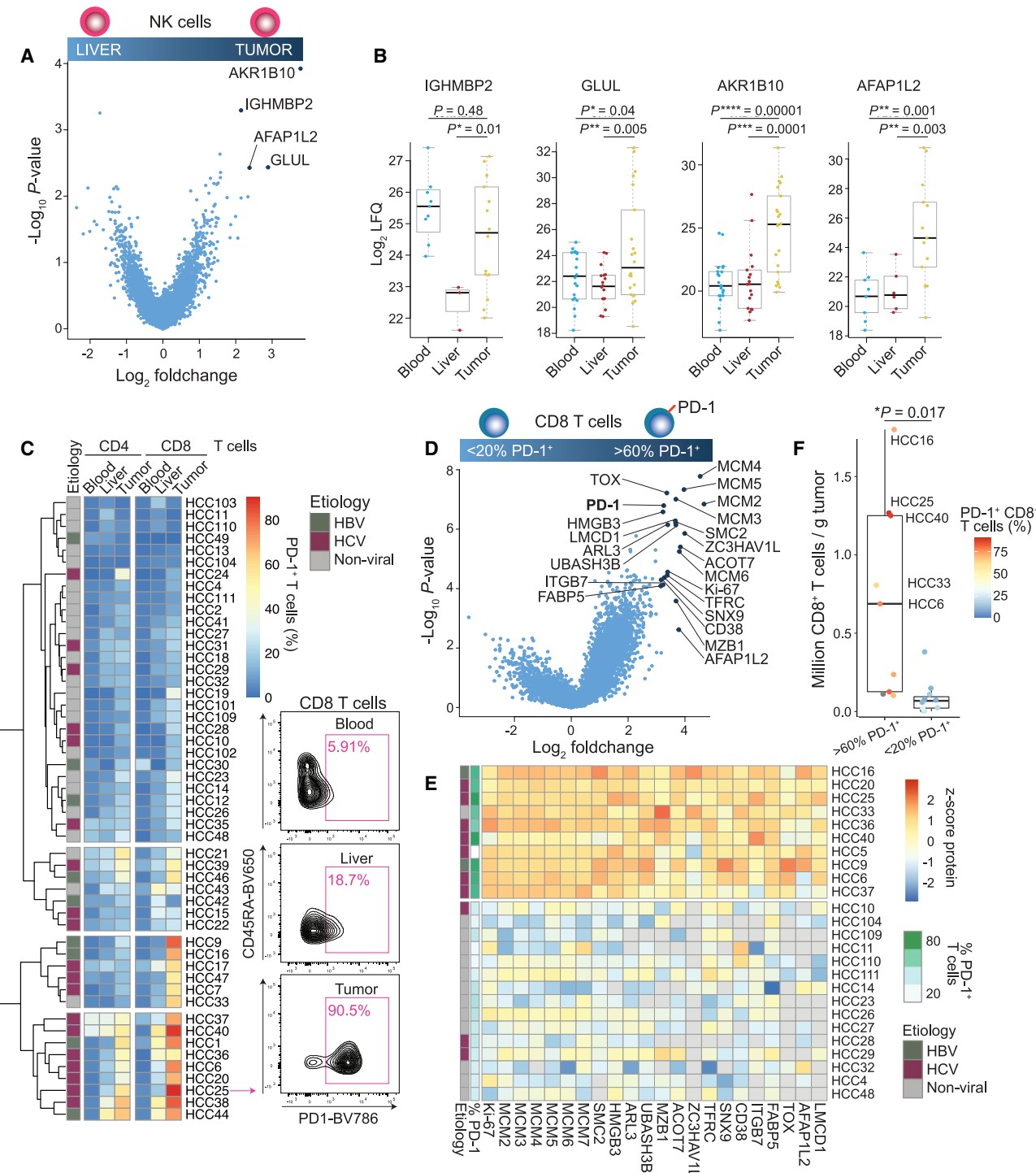

**Cell Genomics**

**Figure 4. PD-1+ T cells in tumors with dysfunctional properties upregulate AFAP1L2**

(A) Volcano plot from differential abundance analysis (two-tailed Welch's t test) between proteomes of NK cells isolated from liver and tumor tissue.

(B) Boxplots showing abundance of selected proteins in NK cells isolated from blood, liver, and tumor tissue. Each dot represents a sample from a different patient.

(C) Heatmap showing the percentage of PD-1 in CD4+ and CD8+ T cells isolated from blood, liver, and tumor tissue of HCC patients as determined by flow cytometry. Contour plots on the right side show an example of the PD-1 staining for patient HCC25.

(D) Volcano plot from differential abundance analysis (two-tailed Welch's t test) between proteomes of CD8+ T cells with a high (>60%, n = 10) and low (<20%, n = 15) percentage of PD-1.

*(legend continued on next page)*

## DISCUSSION

While current immunotherapies have shown clinical activity in HCC patients, the majority of patients fail to respond.[17] Therefore, it is important to identify molecular mechanisms that contribute to immunosuppression in tumors. In this study, we generated a comprehensive proteomics dataset on cancer cells, macrophages, NK cells, and T cells from an HCC patient cohort. Our data captured known aspects of tumor immunology and provided new insights into the behavior of macrophages NK cells and T cells in tumors.

We used mass spectrometry-based proteomics to profile fluorescence-activated cell sorting (FACS)-sorted immune cells isolated from blood, liver tissue, and tumors. During the isolation procedure, tissues are mechanically dissociated and enzymatically digested, which can lead to the degradation of mRNAs and to a lesser extent of proteins. While RNA degradation can lead to artifactual changes in transcriptomes, protein degradation is minimal and has little impact on proteomic measurements, with the exception of enzymatically digested membrane proteins.[49] While measuring transcripts is typically used as a proxy for protein abundance, MS allows the direct quantification of proteins, including those that are not synthesized in a cell but taken up from the surrounding environment.

Our proteome data showed that liver tumors largely alter the phenotype of macrophages. Similar observations were previously made in liver, endometrial, breast, and lung cancer by employing transcriptomics.[50–52] We found that tumor macrophages in HCC displayed the GPI-anchored protein THY-1 on their cell surface. Tumor macrophages did not express *THY-1* mRNA but acquired the protein from the HCC tumor microenvironment. Interestingly, a study in a breast cancer model showed that THY-1 is upregulated in cancer stem cells (CSCs) and mediates physical interactions with tumor macrophages. This interaction allows tumor macrophages to create a CSC niche.[53] THY-1 is also expressed on activated endothelial cells, where it mediates the adhesion of monocytes via its counter receptor CD11B (MAC1).[54] It is possible that during these interactions, macrophages extract THY-1 from the membrane of endothelial cells or CSCs, thus shaping the macrophage phenotype within the TME.

In TAMs isolated from liver tumors, SGPL1 protein was strongly enriched. Inspection of published single-cell RNA-seq data of macrophages isolated from HCC tumors[52] showed that SGPL1 mRNA was present in tumor macrophages but was not as strongly enriched as in our proteome dataset. SGPL1 degrades the signaling sphingolipid S1P in the membrane of the endoplasmic reticulum and is crucial for S1P homeostasis. S1P can be exported from cells, where it interacts with S1P receptors (S1PRs). S1PRs mediate a broad range of cellular functions, including trafficking of lymphocytes. For example, the egress of lymphocytes from lymphoid organs is dependent on S1P receptor 1 (S1PR1). Lymphocytes expressing S1PR1 migrate toward S1P, whose concentration is low in lymphoid organs but high in the blood and lymphatics.[55] In *Sgpl1*$^{-/-}$ mice, S1P concentrations in lymphoid organs are increased, which inhibits the egress of lymphocytes from lymphoid organs, causing a decrease in the number of circulating lymphocytes.[56] Extracellular and intracellular S1P can activate the NF-kB pathway, which inhibits apoptosis in myeloid cells and enhances cytokine production.[57] For example, *Sgpl1*$^{-/-}$ mice exhibit higher concentrations of inflammatory cytokines in their sera in response to an LPS challenge.[58]

In line with an inflammatory phenotype of *Sgpl1*$^{-/-}$ mice, we found that BMDMs in which we ablated *Sgpl1* produced more of the inflammatory cytokines IL-6 and IL-12. The resultant inflammatory phenotype may be a consequence of accumulating S1P, which either acts on S1PRs or through intracellular signaling pathways.[59] Strikingly, *Sgpl1*-deficient BMDMs had improved anti-tumor activity, suggesting that SGPL1 and S1P metabolism are potential therapeutic targets to improve anti-tumor immunity. Systemic targeting of SGPL1 may likely cause side effects, given that *Sgpl1*$^{-/-}$ mice show immunological alterations and have a reduced lifespan.[58] As such, a safer approach might be to target SGPL1 specifically in tumor macrophages, for example, by using macrophage-targeting nanoparticles.[60]

Our analyses of T cell proteomes in HCC revealed that AFAP1L2 was uniquely expressed in CD8 TILs in a subset of patients with high percentages of PD1$^+$ TILs, indicative of highly stimulated and proliferative T cells within the TME. These cells also expressed several inhibitory receptors and CD39, and therefore most likely recognize tumor antigens.[13,61,62] Mechanistically, we validated that AFAP1L2 is induced in CD8$^+$ T cells only upon repeated triggering of the TCR. Genetic ablation of *Afap1l2* in murine CD8$^+$ T cells improved their survival and anti-tumor activity and had synergistic effects with PD-L1 blocking antibodies in the clearance of tumors. This suggests that AFAP1L2 is a potential target for T cell-based cancer immunotherapies to treat patients with HCC and other types of tumors.

Since AFAP1L2 reduces T cell survival and effector functions, it operates as a checkpoint to blunt the escalation of immune responses. As AFAP1L2 is expressed only in chronically stimulated T cells, its ablation does not interfere with T cell activation, and thus it is a promising target specific to dysfunctional T cells. Being an intracellular scaffold protein, AFAP1L2 is not accessible to blocking antibodies. Therefore, AFAP1L2 must be targeted through different strategies. For example, it could be targeted systemically with small molecule degraders, which harness the ubiquitin proteasome system to selectively target intracellular proteins.[63] Since adult *Afap1l2*$^{-/-}$ mice do not show any significant anatomical and physiological phenotypes,[64] it is likely that systemic degradation of AFAP1L2 would have limited toxicity.

An additional therapeutic application of our findings would be to genetically ablate *AFAP1L2* in CAR T cells to improve

---

(E) Heatmap showing the relative abundance (*Z* score) of selected proteins in CD8$^+$ T cell proteome samples of individual patients based on the differential analysis in (D).

(F) Number of CD8$^+$ T cells that were isolated and FACS-purified from HCC tumors compared in (D). Dots are color-coded according to the percentage of PD-1$^+$ T cells as determined in (C). The p value is indicated in the graph and was determined using a two-tailed t test.

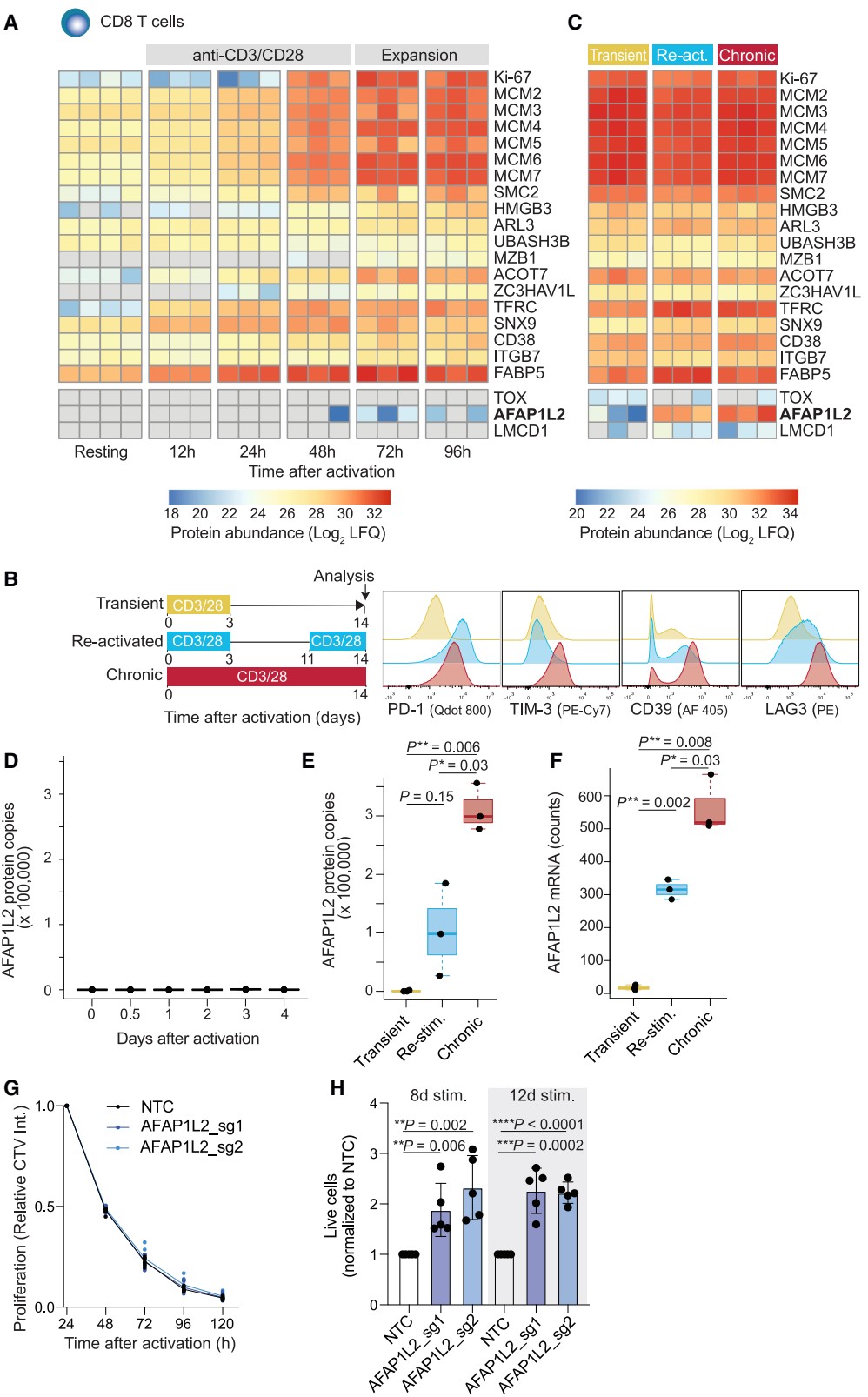

(legend on next page)

their anti-tumor function. Such interventions are crucial for next-generation CAR T cells, because T cell dysfunction is a major contributor to ineffective CAR T cell therapy in solid tumors.[65,66] CAR T cell dysfunction has recently been associated with a transition toward an NK-like phenotype.[67] Consistent with this notion, we found that AFAP1L2 is only expressed in chronically stimulated T cells and NK cells. Whether AFAP1L2 has similar functions in NK cells remains to be determined.

AFAP1L2 is a scaffold protein that can bind to c-Src-containing proteins and PI3K through its proline-rich domain, and it can bind to lipids within membranes through its two pleckstrin homology domains.[37] The 818-amino-acid-long protein also contains a coiled-coil domain with unknown function. In general, scaffold proteins dynamically interact with different partners and shape cell behavior.[68] Notably, the functions of AFAP1L2 are cell type specific since it curtails the survival of chronically stimulated T cells but has the opposite effect in cancer cells. Previous reports showed that silencing of *AFAP1L2* in thyroid and lung cancer cells inhibits cell-cycle progression and survival.[69,70] Consistent with these observations, we found that ablation of *AFAP1L2* in renal (CAKI1) and pancreatic (CAPAN-2) cancer cell lines inhibited their growth (data not shown). Thus, systemic targeting of AFAP1L2 would potentially enhance the activity of chronically stimulated T cells and directly inhibit the growth of cancer cells. How AFAP1L2 controls survival and growth in different cell types remains to be determined.

In summary, our study provides a rich dataset on cancer and immune cell proteomes across a relatively large cohort of patients with HCC. Our findings uncovered new mechanisms underlying T cell dysfunction and mechanisms by which macrophages impede tumor control. These insights are critical for advancing cancer immunotherapy.

### Limitations of the study

In this study, surgically resected liver and tumor tissue was entirely used for immune cell extraction, and no histopathological images were taken to exclude the possibility of microtumors in samples designated as non-tumorous. To confirm correct classification of tumorous and non-tumorous tissue, we relied on the analysis of their proteomes, which showed that they had a clearly distinct profile. However, since we did not assess the ratio of cancer cells to tumor stroma in patient samples,

some differences between tumorous and non-tumorous tissue might be masked.

## STAR★METHODS

Detailed methods are provided in the online version of this paper and include the following:

- KEY RESOURCES TABLE
- RESOURCE AVAILABILITY
  - Lead contact
  - Materials availability
  - Data and code availability
- EXPERIMENTAL MODEL AND STUDY PARTICIPANT DETAILS
  - Human specimens
  - Patient samples
  - Mice
- METHOD DETAILS
  - High resolution mass spectrometry
  - *In vitro* chronic stimulation of human CD8+ T cells
  - RNAseq of chronically stimulated T cells and human CD14+ cells
  - Isolation of OT-I T cells
  - CRISPR-Cas9 targeting of genes of interest in T cells
  - Gene targeting in bone marrow-derived macrophages
  - *In vitro* generation of M1 and M2-polarized human macrophages
  - Co-culture of HCC tumor tissue and CD14+ monocytes
  - Mouse tumor models
  - Flow cytometry
- QUANTIFICATION AND STATISTICAL ANALYSIS
- ADDITIONAL RESOURCES

### SUPPLEMENTAL INFORMATION

### ACKNOWLEDGMENTS

We thank David Jarrossay for cell sorting and Tanja Rezzonico Jost for help with mouse experiments. R.G. is supported in part by the European Research

---

**Figure 5. AFAP1L2 is upregulated in chronically activated T cells and blunts their survival**

(A) FACS-purified naive CD8+ T cells from four healthy donors were either analyzed directly by LC-MS or after increasing times of activation with antibodies to CD3 and CD28. Heatmap shows the abundance of proteins that were significantly upregulated in tumor-infiltrating CD8+ T cells with a high percentage of PD-1+ cells (see Figure 4). Each column represents a different donor.

(B) Schematic of the activation conditions of naive CD8+ T cells and flow cytometry plots showing the expression of PD-1, TIM-3, CD39, and LAG-3. See also Figure S4B for quantifications.

(C) Naive CD8+ T cells were activated as indicated in the schematic in (B) and then analyzed by LC-MS. Heatmap shows the abundance of indicated proteins, where each column represents a different donor.

(D) Plot shows AFAP1L2 protein copy numbers at different time points following activation. n = 4 for resting T cells, and n = 3 for other time points.

(E) AFAP1L2 protein copy numbers in CD8+ T cells following different activation schemes. n = 3. Two-tailed t test.

(F) Same as in (E) but T cells were analyzed by RNA-seq, and *AFAP1L2* mRNA counts are shown.

(G) NTC and *AFAP1L2*-edited CD8+ T cells were labeled with CTV and then stimulated with ImmunoCult CD3/CD28/CD2 Activator (1:333). Proliferation (CTV intensity) was followed by flow cytometry over the time course of 5 days. n = 9 from three different donors.

(H) NTC and *AFAP1L2*-edited CD8+ T cells were stimulated with plate-bound anti-CD3/CD28 antibodies for 8 and 12 days, and the number of surviving T cells was quantified by flow cytometry. n = 5 from five donors. Two-tailed t test.

**Cell Genomics**

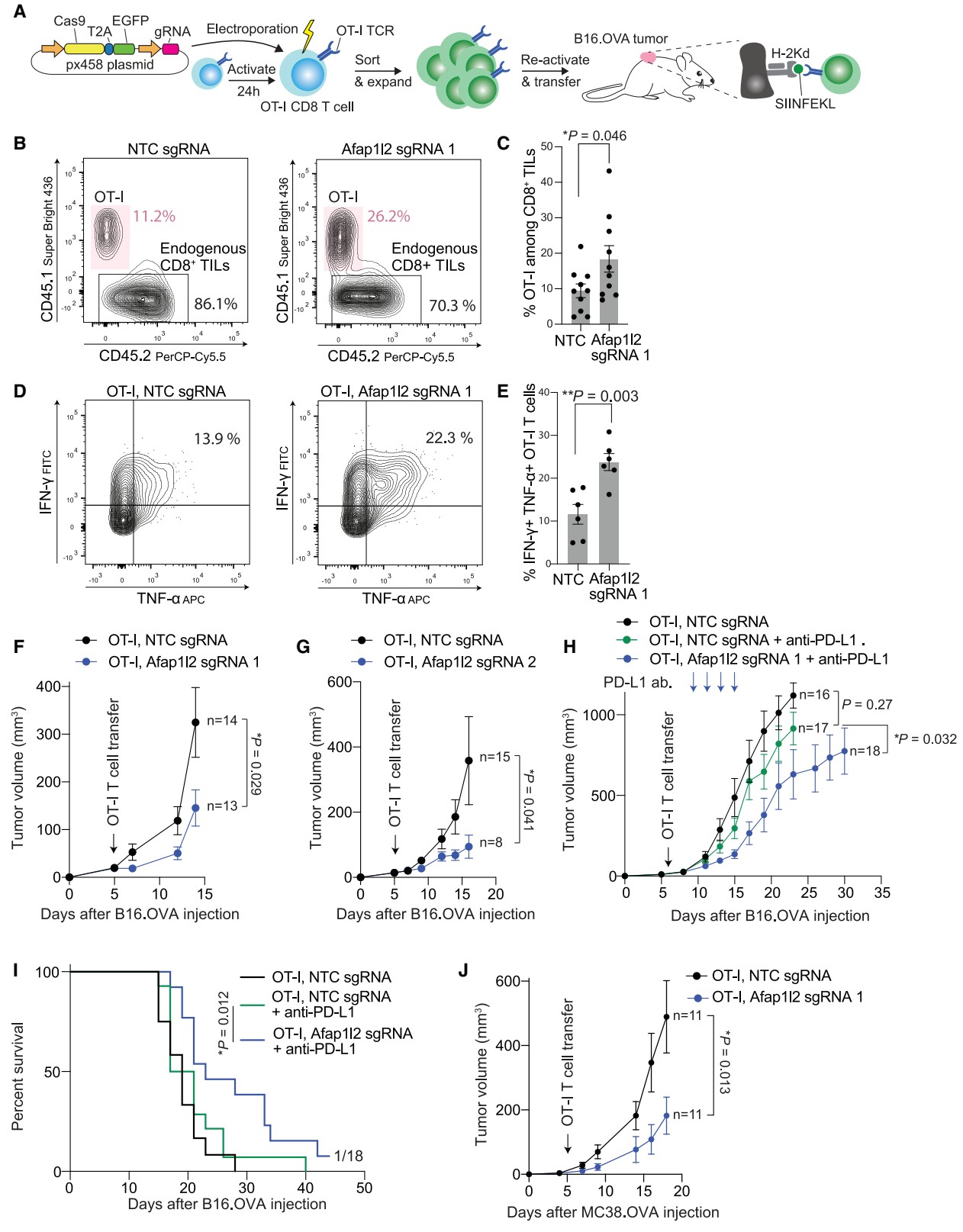

**CellPress**

Council (803150), by Swiss Cancer Research (KFS-4593-08-2018), and by Roche through their postdoctoral fellowship program.

## AUTHOR CONTRIBUTIONS

Conceptualization: R.G. and F.P.C. Methodology: R.G. and F.P.C. Investigation: F.P.C., J.N., J.v.R., M.P., T.W., W.J., G.Z., G.A., G.C., and X.Z. Resources: E.L., G.L.B., C.V.C., A.S., L.S., C.S., S.G., S.d.D., A.C., E.B., N.R., G.E., R.M., and L.T.J. Writing: R.G., I.V., and F.P.C. Visualization: R.G. Supervision: R.G., H.M., N.P., T.N., and M.F. Funding acquisition: R.G.

## DECLARATION OF INTERESTS

The Geiger laboratory received funding from F. Hoffmann-La Roche AG for this study. M.F., T.N., H.M., and N.P. were or are employees and shareholders of F. Hoffmann-La Roche AG. R.G. is a co-founder of Encentrio Therapeutics and a member of the scientific executive board. L.T.J. is a co-founder and former board member of Cimeio Therapeutics AG.

## INCLUSION AND DIVERSITY

We support inclusive, diverse, and equitable conduct of research.

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

---

**Figure 6. T cells devoid of Afap1l2 have superior anti-tumor activity**

(A) Schematic showing the workflow to analyze the impact of gene editing on the anti-tumor activity of T cells.

(B and C) $5 \times 10^5$ B16.OVA cells were subcutaneously injected into C57BL/6 mice. 5 days later, $10^6$ control or *Afap1l2*-edited OT-I T cells were adoptively transferred into mice. 5 days later, tumors were disaggregated, and the number of OT-I T cells and endogenous T cells was quantified by flow cytometry. A representative plot is shown in (B), and quantifications from 10 mice are shown in (C). P value is shown on the graph and was determined using a two-tailed t test. Bars represent SEM throughout.

(D and E) Same as in (B) and (C) but isolated OT-I T cells were re-stimulated with PMA/Ionomycin, and intracellular cytokines were analyzed by flow cytometry. A representative plot is shown in (D), and quantifications from six mice are shown in (E).

(F) $5 \times 10^5$ B16.OVA cells were subcutaneously injected into C57BL/6 mice. 6 days later, $10^6$ control or *Afap1l2*-edited OT-I T cells (sgRNA 1) were adoptively transferred into mice, and the size of tumors was followed. P value was determined by two-way ANOVA and is shown on the graph together with numbers of mice.

(G) Same as in (F) but *Afap1l2* was edited with a different sgRNA.

(H and I) $5 \times 10^5$ B16.OVA cells were subcutaneously injected into C57BL/6 mice. 6 days later, $10^6$ control or *Afap1l2*-edited OT-I T cells were adoptively transferred into mice. Mice received four subcutaneous injections of anti-PD-L1 antibodies indicated by arrows on the graph. Tumor growth curves are shown in (H) and survival curves in (I). P values in (H) were determined by ANOVA and in (I) by Mantel-Cox log rank test.

(J) $5 \times 10^5$ MC38.OVA cells were subcutaneously injected into $Cd3e^{-/-}$ mice. 5 days later, $10^6$ control or *Afap1l2*-edited OT-I T cells were adoptively transferred into mice, and the size of tumors was followed. P value was determined by two-way ANOVA and is shown on the graph together with numbers of mice.

for hepatocellular carcinoma. Nat. Rev. Clin. Oncol. *19*, 151–172. https://doi.org/10.1038/s41571-021-00573-2.

18. Hou, J., Zhang, H., Sun, B., and Karin, M. (2020). The immunobiology of hepatocellular carcinoma in humans and mice: basic concepts and therapeutic implications. J. Hepatol. *72*, 167–182. https://doi.org/10.1016/j.jhep.2019.08.014.

19. Iannacone, M., Andreata, F., and Guidotti, L.G. (2022). Immunological insights in the treatment of chronic hepatitis B. Curr. Opin. Immunol. *77*, 102207. https://doi.org/10.1016/j.coi.2022.102207.

20. Li, X., Ramadori, P., Pfister, D., Seehawer, M., Zender, L., and Heikenwalder, M. (2021). The immunological and metabolic landscape in primary and metastatic liver cancer. Nat. Rev. Cancer *21*, 541–557. https://doi.org/10.1038/s41568-021-00383-9.

21. Protzer, U., Maini, M.K., and Knolle, P.A. (2012). Living in the liver: hepatic infections. Nat. Rev. Immunol. *12*, 201–213. https://doi.org/10.1038/nri3169.

22. Jiang, Y., Sun, A., Zhao, Y., Ying, W., Sun, H., Yang, X., Xing, B., Sun, W., Ren, L., Hu, B., et al. (2019). Proteomics identifies new therapeutic targets of early-stage hepatocellular carcinoma. Nature *567*, 257–261. https://doi.org/10.1038/s41586-019-0987-8.

23. Tang, L., Zeng, J., Geng, P., Fang, C., Wang, Y., Sun, M., Wang, C., Wang, J., Yin, P., Hu, C., et al. (2018). Global metabolic profiling identifies a pivotal role of proline and hydroxyproline metabolism in supporting hypoxic response in hepatocellular carcinoma. Clin. Cancer Res. *24*, 474–485. https://doi.org/10.1158/1078-0432.CCR-17-1707.

24. Nwosu, Z.C., Megger, D.A., Hammad, S., Sitek, B., Roessler, S., Ebert, M.P., Meyer, C., and Dooley, S. (2017). Identification of the consistently altered metabolic targets in human hepatocellular carcinoma. Cell. Mol. Gastroenterol. Hepatol. *4*, 303–323.e1. https://doi.org/10.1016/j.jcmgh.2017.05.004.

25. Geiger, R., Rieckmann, J.C., Wolf, T., Basso, C., Feng, Y., Fuhrer, T., Kogadeeva, M., Picotti, P., Meissner, F., Mann, M., et al. (2016). L-arginine modulates T cell metabolism and enhances survival and anti-tumor activity. Cell *167*, 829–842.e13. https://doi.org/10.1016/j.cell.2016.09.031.

26. Canale, F.P., Ramello, M.C., Núñez, N., Araujo Furlan, C.L., Bossio, S.N., Gorosito Serrán, M., Tosello Boari, J., Del Castillo, A., Ledesma, M., Sedlik, C., et al. (2018). CD39 expression defines cell exhaustion in tumor-infiltrating CD8+ T cells. Cancer Res. *78*, 115–128. https://doi.org/10.1158/0008-5472.CAN-16-2684.

27. Lloyd, A.F., Davies, C.L., Holloway, R.K., Labrak, Y., Ireland, G., Carradori, D., Dillenburg, A., Borger, E., Soong, D., Richardson, J.C., et al. (2019). Central nervous system regeneration is driven by microglia necroptosis and repopulation. Nat. Neurosci. *22*, 1046–1052. https://doi.org/10.1038/s41593-019-0418-z.

28. Wolf, T., Jin, W., Zoppi, G., Vogel, I.A., Akhmedov, M., Bleck, C.K.E., Beltraminelli, T., Rieckmann, J.C., Ramirez, N.J., Benevento, M., et al. (2020). Dynamics in protein translation sustaining T cell preparedness. Nat. Immunol. *21*, 927–937. https://doi.org/10.1038/s41590-020-0714-5.

29. Liu, L.-Z., Zhang, Z., Zheng, B.-H., Shi, Y., Duan, M., Ma, L.J., Wang, Z.C., Dong, L.Q., Dong, P.P., Shi, J.Y., et al. (2019). CCL15 recruits suppressive monocytes to facilitate immune escape and disease progression in hepatocellular carcinoma. Hepatology *69*, 143–159. https://doi.org/10.1002/hep.30134.

30. DiStefano, J.K., and Davis, B. (2019). Diagnostic and prognostic potential of AKR1B10 in human hepatocellular carcinoma. Cancers *11*, 486. https://doi.org/10.3390/cancers11040486.

31. Sauzay, C., Voutetakis, K., Chatziioannou, A., Chevet, E., and Avril, T. (2019). CD90/Thy-1, a cancer-associated cell surface signaling molecule. Front. Cell Dev. Biol. *7*, 66. https://doi.org/10.3389/fcell.2019.00066.

32. Palmieri, E.M., Menga, A., Martín-Pérez, R., Quinto, A., Riera-Domingo, C., De Tullio, G., Hooper, D.C., Lamers, W.H., Ghesquière, B., McVicar,

D.W., et al. (2017). Pharmacologic or genetic targeting of glutamine synthetase skews macrophages toward an M1-like phenotype and inhibits tumor metastasis. Cell Rep. *20*, 1654–1666. https://doi.org/10.1016/j.celrep.2017.07.054.

33. Marigo, I., Trovato, R., Hofer, F., Ingangi, V., Desantis, G., Leone, K., De Sanctis, F., Ugel, S., Canè, S., Simonelli, A., et al. (2020). Disabled homolog 2 controls prometastatic activity of tumor-associated MacrophagesDAB2-expressing TAMs promote cancer cell invasion. Cancer Discov. *10*, 1758–1773.

34. Weigert, A., Olesch, C., and Brüne, B. (2019). Sphingosine-1-Phosphate and macrophage biology-how the sphinx tames the big eater. Front. Immunol. *10*, 1706. https://doi.org/10.3389/fimmu.2019.01706.

35. Brinkman, E.K., Chen, T., Amendola, M., and van Steensel, B. (2014). Easy quantitative assessment of genome editing by sequence trace decomposition. Nucleic Acids Res. *42*, e168. https://doi.org/10.1093/nar/gku936.

36. Brunda, M.J., Luistro, L., Warrier, R.R., Wright, R.B., Hubbard, B.R., Murphy, M., Wolf, S.F., and Gately, M.K. (1993). Antitumor and antimetastatic activity of interleukin 12 against murine tumors. J. Exp. Med. *178*, 1223–1230. https://doi.org/10.1084/jem.178.4.1223.

37. Xu, J., Bai, X.-H., Lodyga, M., Han, B., Xiao, H., Keshavjee, S., Hu, J., Zhang, H., Yang, B.B., and Liu, M. (2007). XB130, a novel adaptor protein for signal transduction. J. Biol. Chem. *282*, 16401–16412. https://doi.org/10.1074/jbc.M701684200.

38. Rieckmann, J.C., Geiger, R., Hornburg, D., Wolf, T., Kveler, K., Jarrossay, D., Sallusto, F., Shen-Orr, S.S., Lanzavecchia, A., Mann, M., and Meissner, F. (2017). Social network architecture of human immune cells unveiled by quantitative proteomics. Nat. Immunol. *18*, 583–593. https://doi.org/10.1038/ni.3693.

39. Alfei, F., Kanev, K., Hofmann, M., Wu, M., Ghoneim, H.E., Roelli, P., Utzschneider, D.T., von Hoesslin, M., Cullen, J.G., Fan, Y., et al. (2019). TOX reinforces the phenotype and longevity of exhausted T cells in chronic viral infection. Nature *571*, 265–269. https://doi.org/10.1038/s41586-019-1326-9.

40. Khan, O., Giles, J.R., McDonald, S., Manne, S., Ngiow, S.F., Patel, K.P., Werner, M.T., Huang, A.C., Alexander, K.A., Wu, J.E., et al. (2019). TOX transcriptionally and epigenetically programs CD8+ T cell exhaustion. Nature *571*, 211–218. https://doi.org/10.1038/s41586-019-1325-x.

41. Scott, A.C., Dündar, F., Zumbo, P., Chandran, S.S., Klebanoff, C.A., Shakiba, M., Trivedi, P., Menocal, L., Appleby, H., Camara, S., et al. (2019). TOX is a critical regulator of tumour-specific T cell differentiation. Nature *571*, 270–274. https://doi.org/10.1038/s41586-019-1324-y.

42. Philip, M., Fairchild, L., Sun, L., Horste, E.L., Camara, S., Shakiba, M., Scott, A.C., Viale, A., Lauer, P., Merghoub, T., et al. (2017). Chromatin states define tumour-specific T cell dysfunction and reprogramming. Nature *545*, 452–456. https://doi.org/10.1038/nature22367.

43. Carpino, N., Turner, S., Mekala, D., Takahashi, Y., Zang, H., Geiger, T.L., Doherty, P., and Ihle, J.N. (2004). Regulation of ZAP-70 activation and TCR signaling by two related proteins, Sts-1 and Sts-2. Immunity *20*, 37–46. https://doi.org/10.1016/s1074-7613(03)00351-0.

44. Zhang, L., Yu, X., Zheng, L., Zhang, Y., Li, Y., Fang, Q., Gao, R., Kang, B., Zhang, Q., Huang, J.Y., et al. (2018). Lineage tracking reveals dynamic relationships of T cells in colorectal cancer. Nature *564*, 268–272. https://doi.org/10.1038/s41586-018-0694-x.

45. Guo, X., Zhang, Y., Zheng, L., Zheng, C., Song, J., Zhang, Q., Kang, B., Liu, Z., Jin, L., Xing, R., et al. (2018). Global characterization of T cells in non-small-cell lung cancer by single-cell sequencing. Nat. Med. *24*, 978–985. https://doi.org/10.1038/s41591-018-0045-3.

46. Zhu, J., Petit, P.-F., and Van den Eynde, B.J. (2019). Apoptosis of tumor-infiltrating T lymphocytes: a new immune checkpoint mechanism. Cancer Immunol. Immunother. *68*, 835–847. https://doi.org/10.1007/s00262-018-2269-y.

47. Kornete, M., Marone, R., and Jeker, L.T. (2018). Highly efficient and versatile plasmid-based gene editing in primary T cells. J. Immunol. *200*, 2489–2501. https://doi.org/10.4049/jimmunol.1701121.

48. Malissen, M., Gillet, A., Ardouin, L., Bouvier, G., Trucy, J., Ferrier, P., Vivier, E., and Malissen, B. (1995). Altered T cell development in mice with a targeted mutation of the CD3-epsilon gene. EMBO J. *14*, 4641–4653. https://doi.org/10.1002/j.1460-2075.1995.tb00146.x.

49. Shao, W., Guo, T., Toussaint, N.C., Xue, P., Wagner, U., Li, L., Charmpi, K., Zhu, Y., Wu, J., Buljan, M., et al. (2019). Comparative analysis of mRNA and protein degradation in prostate tissues indicates high stability of proteins. Nat. Commun. *10*, 2524. https://doi.org/10.1038/s41467-019-10513-5.

50. Cassetta, L., Fragkogianni, S., Sims, A.H., Swierczak, A., Forrester, L.M., Zhang, H., Soong, D.Y.H., Cotechini, T., Anur, P., Lin, E.Y., et al. (2019). Human tumor-associated macrophage and monocyte transcriptional landscapes reveal cancer-specific reprogramming, biomarkers, and therapeutic targets. Cancer Cell *35*, 588–602.e10. https://doi.org/10.1016/j.ccell.2019.02.009.

51. Lavin, Y., Kobayashi, S., Leader, A., Amir, E.A.D., Elefant, N., Bigenwald, C., Remark, R., Sweeney, R., Becker, C.D., Levine, J.H., et al. (2017). Innate immune landscape in early lung adenocarcinoma by paired single-cell analyses. Cell *169*, 750–765.e17. https://doi.org/10.1016/j.cell.2017.04.014.

52. Zhang, Q., He, Y., Luo, N., Patel, S.J., Han, Y., Gao, R., Modak, M., Carotta, S., Haslinger, C., Kind, D., et al. (2019). Landscape and dynamics of single immune cells in hepatocellular carcinoma. Cell *179*, 829–845.e20. https://doi.org/10.1016/j.cell.2019.10.003.

53. Lu, H., Clauser, K.R., Tam, W.L., Fröse, J., Ye, X., Eaton, E.N., Reinhardt, F., Donnenberg, V.S., Bhargava, R., Carr, S.A., and Weinberg, R.A. (2014). A breast cancer stem cell niche supported by juxtacrine signalling from monocytes and macrophages. Nat. Cell Biol. *16*, 1105–1117. https://doi.org/10.1038/ncb3041.

54. Wetzel, A., Chavakis, T., Preissner, K.T., Sticherling, M., Haustein, U.F., Anderegg, U., and Saalbach, A. (2004). Human Thy-1 (CD90) on activated endothelial cells is a counterreceptor for the leukocyte integrin Mac-1 (CD11b/CD18). J. Immunol. *172*, 3850–3859. https://doi.org/10.4049/jimmunol.172.6.3850.

55. Matloubian, M., Lo, C.G., Cinamon, G., Lesneski, M.J., Xu, Y., Brinkmann, V., Allende, M.L., Proia, R.L., and Cyster, J.G. (2004). Lymphocyte egress from thymus and peripheral lymphoid organs is dependent on S1P receptor 1. Nature *427*, 355–360. https://doi.org/10.1038/nature02284.

56. Weber, C., Krueger, A., Münk, A., Bode, C., Van Veldhoven, P.P., and Gräler, M.H. (2009). Discontinued postnatal thymocyte development in sphingosine 1-phosphate-lyase-deficient mice. J. Immunol. *183*, 4292–4301. https://doi.org/10.4049/jimmunol.0901724.

57. Spiegel, S., and Milstien, S. (2011). The outs and the ins of sphingosine-1-phosphate in immunity. Nat. Rev. Immunol. *11*, 403–415. https://doi.org/10.1038/nri2974.

58. Allende, M.L., Bektas, M., Lee, B.G., Bonifacino, E., Kang, J., Tuymetova, G., Chen, W., Saba, J.D., and Proia, R.L. (2011). Sphingosine-1-phosphate lyase deficiency produces a pro-inflammatory response while impairing neutrophil trafficking. J. Biol. Chem. *286*, 7348–7358. https://doi.org/10.1074/jbc.M110.171819.

59. Strub, G.M., Maceyka, M., Hait, N.C., Milstien, S., and Spiegel, S. (2010). Extracellular and intracellular actions of sphingosine-1-phosphate. Sphingolipids as Signaling and Regulatory Molecules, pp. 141–155.

60. Hu, G., Guo, M., Xu, J., Wu, F., Fan, J., Huang, Q., Yang, G., Lv, Z., Wang, X., and Jin, Y. (2019). Nanoparticles targeting macrophages as potential clinical therapeutic agents against cancer and inflammation. Front. Immunol. *10*, 1998. https://doi.org/10.3389/fimmu.2019.01998.

61. Gros, A., Robbins, P.F., Yao, X., Li, Y.F., Turcotte, S., Tran, E., Wunderlich, J.R., Mixon, A., Farid, S., Dudley, M.E., et al. (2014). PD-1

identifies the patient-specific CD8+ tumor-reactive repertoire infiltrating human tumors. J. Clin. Invest. *124*, 2246–2259. https://doi.org/10.1172/JCI73639.

62. Simoni, Y., Becht, E., Fehlings, M., Loh, C.Y., Koo, S.L., Teng, K.W.W., Yeong, J.P.S., Nahar, R., Zhang, T., Kared, H., et al. (2018). Bystander CD8+ T cells are abundant and phenotypically distinct in human tumour infiltrates. Nature *557*, 575–579. https://doi.org/10.1038/s41586-018-0130-2.

63. Békés, M., Langley, D.R., and Crews, C.M. (2022). PROTAC targeted protein degraders: the past is prologue. Nat. Rev. Drug Discov. *21*, 181–200. https://doi.org/10.1038/s41573-021-00371-6.

64. Zhao, J., Wang, Y., Wakeham, A., Hao, Z., Toba, H., Bai, X., Keshavjee, S., Mak, T.W., and Liu, M. (2014). XB130 deficiency affects tracheal epithelial differentiation during airway repair. PLoS One *9*, e108952. https://doi.org/10.1371/journal.pone.0108952.

65. Poorebrahim, M., Melief, J., Pico de Coaña, Y., L Wickström, S., Cid-Arregui, A., and Kiessling, R. (2021). Counteracting CAR T cell dysfunction. Oncogene *40*, 421–435. https://doi.org/10.1038/s41388-020-01501-x.

66. Young, R.M., Engel, N.W., Uslu, U., Wellhausen, N., and June, C.H. (2022). Next-generation CAR T-cell therapies. Cancer Discov. *12*, 1625–1633. https://doi.org/10.1158/2159-8290.CD-21-1683.

67. Good, C.R., Aznar, M.A., Kuramitsu, S., Samareh, P., Agarwal, S., Donahue, G., Ishiyama, K., Wellhausen, N., Rennels, A.K., Ma, Y., et al. (2021). An NK-like CAR T cell transition in CAR T cell dysfunction. Cell *184*, 6081–6100.e26. https://doi.org/10.1016/j.cell.2021.11.016.

68. Good, M.C., Zalatan, J.G., and Lim, W.A. (2011). Scaffold proteins: hubs for controlling the flow of cellular information. Science *332*, 680–686. https://doi.org/10.1126/science.1198701.

69. Lodyga, M., De Falco, V., Bai, X.H., Kapus, A., Melillo, R.M., Santoro, M., and Liu, M. (2009). XB130, a tissue-specific adaptor protein that couples the RET/PTC oncogenic kinase to PI 3-kinase pathway. Oncogene *28*, 937–949. https://doi.org/10.1038/onc.2008.447.

70. Shiozaki, A., Lodyga, M., Bai, X.-H., Nadesalingam, J., Oyaizu, T., Winer, D., Asa, S.L., Keshavjee, S., and Liu, M. (2011). XB130, a novel adaptor protein, promotes thyroid tumor growth. Am. J. Pathol. *178*, 391–401. https://doi.org/10.1016/j.ajpath.2010.11.024.

71. Lanzavecchia, A., and Scheidegger, D. (1987). The use of hybrid hybridomas to target human cytotoxic T lymphocytes. Eur. J. Immunol. *17*, 105–111. https://doi.org/10.1002/eji.1830170118.

72. Bellone, M., Cantarella, D., Castiglioni, P., Crosti, M.C., Ronchetti, A., Moro, M., Garancini, M.P., Casorati, G., and Dellabona, P. (2000). Relevance of the tumor antigen in the validation of three vaccination strategies for melanoma. J. Immunol. *165*, 2651–2656. https://doi.org/10.4049/jimmunol.165.5.2651.

73. Cox, J., and Mann, M. (2008). MaxQuant enables high peptide identification rates, individualized p.p.b.-range mass accuracies and proteome-wide protein quantification. Nat. Biotechnol. *26*, 1367–1372. https://doi.org/10.1038/nbt.1511.

74. Rappsilber, J., Mann, M., and Ishihama, Y. (2007). Protocol for micro-purification, enrichment, pre-fractionation and storage of peptides for proteomics using StageTips. Nat. Protoc. *2*, 1896–1906. https://doi.org/10.1038/nprot.2007.261.

75. Cox, J., and Mann, M. (2008). MaxQuant enables high peptide identification rates, individualized p.p.b.-range mass accuracies and proteome-wide protein quantification. Nat. Biotechnol. *26*, 1367–1372. https://doi.org/10.1038/nbt.1511.

76. Cox, J., Neuhauser, N., Michalski, A., Scheltema, R.A., Olsen, J.V., and Mann, M. (2011). Andromeda: a peptide search engine integrated into the MaxQuant environment. J. Proteome Res. *10*, 1794–1805. https://doi.org/10.1021/pr101065j.

77. Cox, J., Hein, M.Y., Luber, C.A., Paron, I., Nagaraj, N., and Mann, M. (2014). Accurate proteome-wide label-free quantification by delayed

**Cell Genomics**

normalization and maximal peptide ratio extraction, termed MaxLFQ. Mol. Cell. Proteomics *13*, 2513–2526. https://doi.org/10.1074/mcp.M113. 031591.

78. Manes, N.P., Calzola, J.M., Kaplan, P.R., Fraser, I.D.C., Germain, R.N., Meier-Schellersheim, M., and Nita-Lazar, A. (2022). Absolute protein quantitation of the mouse macrophage Toll-like receptor and chemotaxis pathways. Sci. Data *9*, 491. https://doi.org/10.1038/s41597-022-01612-y.

79. Dölz, M., Marone, R., and Jeker, L.T. (2021). Plasmid-or ribonucleoprotein-mediated CRISPR/cas gene editing in primary murine T cells. In T-Helper Cells (Springer), pp. 255–264.

80. Zarif, J.C., Hernandez, J.R., Verdone, J.E., Campbell, S.P., Drake, C.G., and Pienta, K.J. (2016). A phased strategy to differentiate human CD14+monocytes into classically and alternatively activated macrophages and dendritic cells. Biotechniques *61*, 33–41. https://doi.org/10. 2144/000114435.

## STAR★METHODS

### KEY RESOURCES TABLE

| REAGENT or RESOURCE | SOURCE | IDENTIFIER |
| --- | --- | --- |
| **Antibodies** | | |
| Human CCR7 BV421 (clone G043H7) | Biolegend | Cat#353208; RRID:AB_11203894 |
| Human CD45RA Qdot™ 655 (clone MEM-56) | ThermoFisher | Cat#Q10069; RRID:AB_2556451 |
| Human CD8 Super Bright 600 (clone RPA-T8) | ThermoFisher | Cat#63-0088-42; RRID:AB_2637470 |
| Human CD4 PE-Cy7 (clone SK3) | BD Biosciences | Cat#557852; RRID:AB_396897 |
| Human CD14 V500 (clone M5E2) | BD Biosciences | Cat#561392; RRID:AB_10611862 |
| Human PD1 BV785 (clone EH12.2H7) | Biolegend | Cat#329930; RRID:AB_2563443 |
| Human CD56 PE-Cy5 (clone N901) | Beckman Coulter | Cat#A07789; RRID:AB_1575976 |
| Human CD19 PE-Texas Red (clone Sj25-C1) | ThermoFisher | Cat#MHCD1917; RRID:AB_10372040 |
| Human CD39 BV421 (clone A1) | Biolegend | Cat#328214; RRID:AB_2564575 |
| Human TIM3 PE-Vio770 (clone REA635) | Miltenyi Biotec | Cat#130-121-334; RRID:AB_2784165 |
| Human CD223 (LAG3) PE (clone 3DS223H) | ThermoFisher | Cat#12-2239-42; RRID:AB_2572597 |
| Human CD3 (clone TR66) | In house | Lanzavecchia and Scheidegger[71] |
| Human CD28 (clone CD28.2) | BD Biosciences | Cat#555725; RRID:AB_396068 |
| Human HLA-DR PE-Cy5 (clone Immu-357) | Beckman Coulter | Cat#A07793 |
| Human CD80 APC (clone 2D10.4) | ThermoFisher | Cat#17-0809-42; RRID:AB_2802217 |
| Human CD86 FITC (clone REA968) | Miltenyi Biotec | Cat#130-116-262; RRID:AB_2727435 |
| Human CD163 APC (clone GHI/61) | ThermoFisher | Cat#17-1639-41; RRID:AB_2573167 |
| Human CD206 PE-Cy7 (clone 19.2) | ThermoFisher | Cat#25-2069-41; RRID:AB_2573425 |
| Human CD45 PerCP-Vio700 (clone REA747) | Miltenyi Biotec | Cat#130-110-774; RRID:AB_2658252 |
| Human CD11b APC-Vio770 (clone REA713) | Miltenyi Biotec | Cat#130-110-614; RRID:AB_2654674 |
| Human THY1 PE (clone 5E10) | ThermoFisher | Cat#12-0909-42; RRID:AB_10670624 |
| Mouse CD3ε (clone 145-2C11) | ThermoFisher | Cat#14-0031-85; RRID:AB_467050 |
| Mouse CD28 (clone 37.51) | ThermoFisher | Cat#14-0281-85; RRID:AB_467191 |
| InVivoPlus anti-mouse PD-L1 (clone 10F.9G2) | Bio X Cell | Cat#BP0101; RRID:AB_10949073 |
| Mouse CD8α Super Bright 702 (clone 53-6.7) | ThermoFisher | Cat#67-0081-82; RRID:AB_2662351 |
| Mouse CD45.1 Super Bright 436 (clone A20) | ThermoFisher | Cat#62-0453-82; RRID:AB_2734990 |
| Mouse CD45.2 PerCP-Cy5.5 (clone 104) | ThermoFisher | Cat#45-0454-82; RRID:AB_953590 |
| Mouse PD1 PE (clone RMP1-30) | ThermoFisher | Cat#12-9981-82; RRID:AB_466290 |
| Mouse CD336 (TIM3) PE-Cy7 (clone RMT3-23) | ThermoFisher | Cat#25-5870-82; RRID:AB_2573483 |
| Mouse CD223 (LAG-3) FITC (clone C9B7W) | ThermoFisher | Cat#11-2231-82; RRID:AB_2572484 |
| Mouse IFNγ FITC (clone XMG1.2) | ThermoFisher | Cat#11-7311-82; RRID:AB_465412 |
| Mouse TNF APC (clone MP6-XT22) | ThermoFisher | Cat#17-7321-82; RRID:AB_469508 |
| Mouse F4/80 eFluor 506 (clone BM8) | ThermoFisher | Cat#48-4801-80; RRID:AB_1548756 |
| Mouse CD86 Super Bright 702 (clone GL1) | ThermoFisher | Cat#67-0862-82; RRID:AB_2717155 |
| Mouse IL6 eFluor 450 (clone MP5-20F3) | ThermoFisher | Cat#48-7061-82; RRID:AB_2574103 |
| Mouse IL-12/IL-23 p40 PE (clone C17.8) | ThermoFisher | Cat#12-7123-82; RRID:AB_466185 |

*(Continued on next page)*

*Continued*

| REAGENT or RESOURCE | SOURCE | IDENTIFIER |
|---|---|---|
| **Biological samples** | | |
| Peripheral blood, liver tissue and tumor tissue samples from Hepatocellular Carcinoma patients. | Department of Medical and Surgical Sciences - DIMEC; Alma Mater Studiorum - Univeristy of Bologna, Bologna, Italy; Morgagni-Pierantoni Hospital, Ausl Romagna, Forlì, Italy (study 750/2016). | N/A |
| Peripheral blood, liver tissue and tumor tissue samples from Hepatocellular Carcinoma patients | Department of Surgery, Medical Faculty Mannheim, Universitätsmedizin Mannheim, Heidelberg University, Mannheim, Germany (study 2012-293N-MA) | N/A |
| Peripheral blood, liver tissue and tumor tissue samples from Hepatocellular Carcinoma patients | Department of Visceral, Thoracic and Vascular Surgery, University Hospital Carl Gustav Carus, University of Technology Dresden, Dresden, Germany (study EK 76032013) | N/A |
| Buffy coats from Healthy Donors as source of primary T cells and monocytes. | Swiss Blood Donation Center of Basel and Lugano, Switzerland. (Authorization 2018-02166/CE 3428) | N/A |
| **Chemicals, peptides, and recombinant proteins** | | |
| RPMI-1640 | Gibco | Cat#42401-018 |
| DMEM | Gibco | Cat#31885-023 |
| McCoy's 5A | ATCC | Cat#30-2007 |
| Phenol Red-free RPMI-1640 | Gibco | Cat#11835-063 |
| EDTA-pH8 (0.5 M) | ITW Reagents | Cat#A3145 |
| Fetal Bovine Serum (FBS) | Gibco | Cat#10270-106 |
| DPBS | Sigma-Aldrich | Cat#D8537 |
| MEM Non-essential amino acids (NEAA) | Gibco | Cat#11140-035 |
| Penicillin-Streptomycin | Gibco | Cat#15070-063 |
| GlutaMAX$^{TM}$ | Gibco | Cat#35050-038 |
| 2-Mercaptoethanol 50 mM | Gibco | Cat#31350-010 |
| HEPESHepes Buffer Solution (1M) | Gibco | Cat#15630-056 |
| Sodium Pyruvate (100 mM) | Gibco | Cat#11360-039 |
| 0.05% Trypsin-EDTA | Gibco | Cat#25300-054 |
| Ficoll-Paque$^{TM}$ PLUS | Cytiva | Cat#17144003 |
| Ammonium bicarbonate | Sigma-Aldrich | Cat#A6141 |
| Urea | Sigma-Aldrich | Cat#51456 |
| Dithiothreitol (DTT) | Sigma-Aldrich | Cat#D0632 |
| Iodoacetamide | Sigma-Aldrich | Cat#I6125 |
| Lysyl EndopeptidaseR (Lys-C) | Fujifilm Wako | Cat#129-02541 |
| Sequencing Grade Modified Trypsin | Promega | Cat#V5111 |
| Water (Optima$^{TM}$ LC/MS) | Fisher Chemical | Cat#11947199(W6-4) |
| Acetonitrile (Optima$^{TM}$ LC/MS) | Fisher Chemical | Cat#10489553 |
| Pierce$^{TM}$ Trifluoroacetic Acid, LC-MS grade | ThermoFisher | Cat#85183 |
| Acetic acid | Merck | Cat#1.00063 |
| Formic Acid, 99% Optima$^{TM}$ LC/MS Grade | Fisher Chemical | Cat#A117-50 |
| PK20 Empore C18 Octadecyl 47mm (C$_{18}$-silica) | Sigma-Aldrich | Cat#66883-U |
| Reprosil-Pur 120 C18-AQ 1.9um (C$_{18}$ particles) | Dr. Maisch GmbH | Cat#r119.aq.0003 |
| NP-40 Surfact-Amps$^{TM}$ Detergent Solution | ThermoFisher | Cat#85124 |
| Tris hydrochloride | Roche | Cat#10812846001 |

*(Continued on next page)*

*Continued*

| REAGENT or RESOURCE | SOURCE | IDENTIFIER |
| --- | --- | --- |
| cOmplete™ ULTRA Tablets | Merck | Cat#5892953001 |
| Human recombinant interleukin-2 (transfected J588L cell supernatant) | In house | N/A |
| Recombinant human interleukin-2 | Sino Biological | Cat#11848-HNAH1-E |
| Zombie Green Fixable Viability Kit | Biolegend | Cat#423111 |
| Zombie UV Fixable Viability Kit | Biolegend | Cat#423107 |
| CellTrace™ Violet Cell Proliferation Kit | ThermoFisher | Cat#C34557 |
| CellTrace™ Far Red Cell Proliferation Kit | ThermoFisher | Cat#C34564 |
| CellTrace™ CFSE Cell Proliferation Kit | ThermoFisher | Cat#C34554 |
| QuickExtract DNA Extraction Solution | Lucigen | Cat#QE0905T |
| PEI MAX™ (MW 40,000) | Polysciences | Cat#24765-100 |
| Recombinant human M-CSF | Sino Biological | Cat#11792-HNAH |
| Recombinant human GM-CSF | Sino Biological | Cat#10015-HNAH |
| Recombinant human IL-4 | Sino Biological | Cat#11846-HNAE |
| Recombinant human IL-13 | Sino Biological | Cat#10369-HNAC |
| Recombinant human IFNγ | Sino Biological | Cat#11725-HNAS |
| LPS-EB Ultrapure | InvivoGen | Cat#tlrl-3pelps |
| ACK Lysing Buffer | Gibco | Cat#A10492-01 |
| Mouse recombinant M-CSF (L929-conditioned media) | In house | N/A |
| Puromycin | InvivoGen | Cat#ant-pr-1 |
| Recombinant mouse IFNγ | Sino Biological | Cat#50709-MNAH |
| Collagenase D | Roche | Cat#11088866001 |
| DNAse I grade II | Roche | Cat#10104159001 |
| Phorbol 12-myristate 13-acetate (PMA) | Sigma-Aldrich | Cat#P1585 |
| Ionomycin | Sigma-Aldrich | Cat#I0634 |
| Brefeldin A | ThermoFisher | Cat#00-4506-51 |
| Monensin | ThermoFisher | Cat#00-4505-51 |
| True-Nuclear™ Transcription Factor Buffer Set | Biolegend | Cat#424401 |
| Critical commercial assays | | |
| Mouse IL-12 p70 DuoSet ELISA kit | R&D Systems | Cat#DY419-05 |
| GenElute™ Mammalian Total RNA Miniprep Kit | Sigma-Aldrich | Cat#RTN70 |
| PrimeFlow™ RNA Assay Kit | ThermoFisher | Cat#88-18005-204 |
| Phusion High-Fidelity PCR Master Mix | ThermoFisher | Cat#F531L |
| EasySep™ Mouse CD8⁺ T Cell Isolation Kit | STEMCELLTechnologies | Cat#19853 |
| Neon Transfection System | ThermoFisher | Cat#MPK5000 |
| Neon 100 uL transfection kit | ThermoFisher | Cat#MPK10096 |
| CD8 MicroBeads | Miltenyi | Cat#130-045-201; RRID:AB_2889920 |
| CD14 MicroBeads | Miltenyi | Cat#130-050-201; RRID:AB_2665482 |
| NucleoSpin Extract II kit | Macherey Nagel | Cat#740609.250 |
| Tumor Dissociation kit | Miltenyi | Cat#130-095-929 |
| Deposited data | | |
| Proteomics data | ProteomeXchange Consortium (PRIDE partner repository) | PXD040957 |
| RNA-seq data CD14⁺ cells from HCC patients | Gene Expression Omnibus | GSE229400 |
| RNA-seq data chronically stimulated T cells | Gene Expression Omnibus | GSE228571 |

*(Continued on next page)*

*Continued*

| REAGENT or RESOURCE | SOURCE | IDENTIFIER |
|---|---|---|
| **Experimental models: Cell lines** | | |
| Mouse: B16.OVA | Dr. Matteo Bellone (San Raffaele Scientific Institute, Milan) | Bellone et al.[72] Original B16 cell line in ATCC: Cat#CRL-6323™ |
| Mouse: MC38.OVA | Dr. Walter Reith (University of Geneva, Geneva) | N/A |
| Mouse: MC38 | Dr. Walter Reith (University of Geneva, Geneva) | N/A |
| Mouse: L929 | ATCC | Cat#CCL-1™ |
| Human: Caki-1 | ATCC | Cat#HTB-46™ |
| Human: Capan-2 | ATCC | Cat#HTB-80™ |
| **Experimental models: Organisms/strains** | | |
| Mouse: C57BL/6 | Charles River | Cat#000664, RRID:IMSR_JAX:000664 |
| Mouse: *Cd3e*−/− C57BL/6 | Malissen et al.[48] | N/A |
| Mouse: ROSA-Cas9 mice: *(B6(C)-Gt(ROSA)26Sorem1.1(CAG-cas9*,-EGFP)Rsky/J)* | The Jackson Laboratory | Cat#028555, RRID:IMSR_JAX:028555 |
| Mouse: *Rag-/-* CD45.1 OT-1: crossing of *B6.129S7-Rag1tm1Mom/J, C57BL/6-Tg(TcraTcrb)1100Mjb/J* and *B6.SJL-Ptprc^a Pepc^b/BoyJ* strains. | Dr. Fabio Grassi (IRB, Bellinzona) | Original strains from Jackson Lab: Cat#002216, RRID:IMSR_JAX:002216 ; Cat#003831, RRID:IMSR_JAX:003831; Cat#002014, RRID:IMSR_JAX:002014 |
| **Oligonucleotides** | | |
| sgRNA and genomic PCR Primers see Table S7 | Microsynth AG | N/A |
| THY1 mRNA detection probe, type 1 | ThermoFisher | Cat#VA1-12482-PF |
| GAPDH mRNA detection probe, type 6 | ThermoFisher | Cat# VA6-10337-PF |
| **Recombinant DNA** | | |
| pSpCas9(BB)-2A-GFP (PX458) | Addgene | Cat#48138; RRID:Addgene_48138 |
| psPAX2 | Addgene | Cat#12260; RRID:Addgene_12260 |
| pMD2.G | Addgene | Cat#12259; RRID:Addgene_12259 |
| lentiCRISPR v2 | Addgene | Cat#52961; RRID:Addgene_103062 |
| pKLV-U6gRNA(BbsI)-PGKpuro-2A-BFP | Addgene | Cat#50946; RRID:Addgene_50946 |
| **Software and algorithms** | | |
| MaxQuant (version 1.6.7.0) | Cox and Mann[73]; http://www.coxdocs.org/doku.php?id=maxquant:start | RRID:SCR_014485 |
| R environment for statistical computing | https://www.r-project.org/ | RRID:SCR_001905 |
| TIDE version 3.3.0 | http://shinyapps.datacurators.nl/tide/ | N/A |
| CrispRGold (version 1.1) | Max-Delbrück-Centrum Für Molekulare Medizin, https://crisprgold.mdc-berlin.de/ | N/A |

## RESOURCE AVAILABILITY

### Lead contact
Further information and requests for resources and reagents should be directed to the Lead Contact, Roger Geiger (roger.geiger@irb.usi.ch).

### Materials availability
This study did not generate new unique reagents.

### Data and code availability

- The mass spectrometry proteomics data have been deposited to the ProteomeXchange Consortium via the PRIDE[1] partner repository with the dataset identifier PXD040957.
- The raw count RNA-seq data have been deposited at GEO and are publicly available as of the date of publication. Accession numbers are GSE229400 for tumor macrophages and GSE228571 for T cells.
- Any additional information required to reanalyze the data reported in this paper is available from the lead contact upon request.

## EXPERIMENTAL MODEL AND STUDY PARTICIPANT DETAILS

### Human specimens

After obtaining written informed consent, peripheral blood, liver and tumor tissue was obtained from HCC patients undergoing liver resection at the Ospedale Sant'Orsola (Bologna), Universitätsklinik (Mannheim) and Universitätsklinikum Carl Gustav Carus (Dresden).

### Patient samples

Fresh tumor and adjacent liver tissue samples were cut into approximately 1 mm$^3$ pieces in RPMI-1640 medium (Invitrogen). Samples were enzymatically digested using the MACS human Tumor Dissociation kit (Miltenyi Biotec) in a gentleMACS$^{TM}$ Dissociator (Miltenyi Biotec) according to manufacturer's instructions (protocol for a sample size of 0.2–1.0 g, h_tumor_01 program). Cell suspensions were filtered through 70 μm Cell-Strainers (BD) in RPMI-1640 medium and then centrifuged at 500 g for 5 min. Cell pellets were re-suspended in 25 mL RPMI-1640 medium. Mononuclear immune cells were separated from tumor and stromal cells by Ficoll gradient centrifugation (15 mL of Ficoll-Histopaque solution; 18°C, 30 min, 774 g). Mononuclear cells and tumor/stroma pellets were washed twice with RPMI-1640. Tumor/stroma cell pellets were washed three times with PBS, snap-frozen in liquid nitrogen and kept at -80°C until processing. Mononuclear immune cells were resuspended in 500 μL MACS buffer (PBS with 2% FBS and 5 mM EDTA), stained and sorted as indicated in Figure S1A. Monocytes/macrophages, and memory T cells were sorted using a FACS ARIA III (BD) as indicated in Figure S1A and recovered in RPMI-1640 supplemented with 1X Glutamax, Penicilin/Streptomycin, Non-essential amino acids and Hepes (GIBCO). Then, cells were washed and resuspended in cold PBS three times before pellets were snap frozen in liquid nitrogen and kept at -80°C.

Peripheral blood was obtained from patients prior to surgery in EDTA tubes. Then, immune cells were obtained by Ficoll gradient centrifugation (5 mL of blood on top of 3 mL of a Ficoll-Histopaque solution). Mononuclear cells were washed and stained for sorting as shown in Figure S1A.

### Mice

Wild type C57BL/6 mice were obtained from Charles River (Italy). CD3e$^{-/-}$ mice were kindly provided by Dr. Bernard Malissen (CIML, Marseille) and have been described previously.[48] *Rag-/- HZ OT-1* mice were obtained by crossing three different strains obtained from Jackson Laboratory: *B6.129S7-Rag1tm1Mom/J*, *C57BL/6 Tg(TcraTcrb)1100Mjb/J* and *B6.SJL-Ptprca Pepcb/BoyJ*. Rosa-Cas9 mice (*B6(C)-Gt(ROSA)26Sorem1.1(CAG-cas9*,-EGFP)Rsky/J)* were from Jackson Laboratory. Mice were maintained under specific pathogen-free conditions in the animal facility of the Institute for Research in Biomedicine (IRB). Five mice per cage were housed in ventilated cages under standardized conditions (20±2ºC, 55±8% relative humidity, 12 h light/dark cycle). Food and water were available *ad libitum*, and mice were examined daily. Female mice were used between 6 and 10 weeks of age. Mice were treated in accordance with the Ticino Cantonal Commission for Animal Welfare, which is in accordance with the Animal Welfare Ordinance and the Animal Experimentation Ordinance from the Swiss Animal Welfare Legislation. Tumor sizes of 1000 mm$^3$ was considered the limit to terminate experiments (cantonal authorization number TI 51/2019).

## METHOD DETAILS

### High resolution mass spectrometry

For tumor/liver stroma, 2-5 mm3 of each pellet were homogenized in 4% SDS in 100 mM Tris pH 7.6 for 10 min at 300 oscillations/minute in a TissueLyser II (Qiagen). Samples were further sonicated in a Bioruptor (Diagenode) (15 cycles, 30s on, 30s off, high mode) and incubated at 95°C for 10 min, after which the lysates were cleared by centrifugation. Proteins were precipitated overnight at −20°C in 80% acetone (VWR), pelleted by centrifugation at 13,000 rpm for 20 min at 4°C and dried on a heating block at 40°C. The protein pellets were re-suspended in 50 mM ammonium bicarbonate buffer (ABC) at pH 8 containing 8M urea (Sigma). For sorted immune cells, pellets were directly re-suspended in 50 mM ABC 8M urea and lysed by Bioruptor sonication. Disulfide bonds were reduced with 10mM DTT (Sigma) and subsequently alkylated with 50 mM iodoacetamide (Sigma). Samples were pre-digested for 2 hours with LysC (Wako Fujifilm, 1:100, w/w) and then diluted 1:4 with 50 mM ABC before trypsin (Promega, 1:100, w/w) was added and the mixtures were incubated overnight at RT. The resulting peptide mixtures were acidified and loaded on C18 StageTips[74]. Peptides were eluted with 80% acetonitrile (ACN), dried using a SpeedVac, and resuspended in 2% ACN, 0.1% trifluoroacetic acid and 0.5% acetic acid for single-shot MS measurement.

Peptides were separated on an EASY-nLC 1200 HPLC system (Thermo Fisher Scientific, Odense) coupled online to a Q Exactive HF mass spectrometer via a nanoelectrospray source (Thermo Fisher Scientific). Peptides were loaded in buffer A (0.1% formic acid) on in house packed columns (75 $\mu$m inner diameter, 50 cm length, and 1.9 $\mu$m C18 particles from Dr. Maisch GmbH) and eluted with a linear 150 min gradient of 5%–30% buffer B (80% ACN, 0.1% formic acid) at a flow rate of 250 nl/min and a column temperature of 50°C. The Q Exactive HF was operated in a data-dependent mode with a survey scan range of 300-1,650 m/z, resolution 60,000 at 200 m/z, maximum injection time 20 ms and AGC target 3e6. Up to the ten most abundant ions with charge 2-5 were isolated with a 1.8 m/z isolation window and subjected to higher-energy collisional dissociation (HCD) fragmentation with a normalized collision energy of 27. MS/MS spectra were acquired with a resolution of 15,000 at 200 m/z, maximum injection time 55 ms and AGC target 1e5. Dynamic exclusion of 30 s was used to reduce repeated sequencing. Data were acquired with the Xcalibur software (Thermo Scientific).

MaxQuant software (version 1.6.7.0) was used to analyze MS raw files.[75] MS/MS spectra were searched against the human Uniprot FASTA database (version June 2019) and a common contaminants database (247 entries) by the Andromeda search engine.[76] Enzyme specificity was set as "Trypsin/P" with a maximum of 2 missed cleavages and 7 as minimum peptide length. N-terminal protein acetylation and methionine oxidation were set as variable modifications, and cysteine carbamidomethylation as a fixed modification. A false discovery rate (FDR) of 1% was required for peptides and proteins. Peptide identification was performed with an allowed initial precursor mass deviation of up to 7 ppm and an allowed fragment mass deviation of 20 ppm. Nonlinear retention time alignment of all measured samples was performed in MaxQuant. Peptide identifications were matched across different replicates within a matching time window of 0.7 min and an alignment time window of 20 min. Protein identification required at least 1 razor peptide. A minimum ratio count of 1 was required for valid quantification events via MaxQuant's Label Free Quantification algorithm.[77] Data were filtered for common contaminants and reverse peptides, and peptides only identified by side modification were excluded from further analysis.

Data analysis was performed using the R statistical computing environment. For statistical analyses, missing values were imputed with a normal distribution of 30% in comparison to the SD of measured values and a 1.8 SD down-shift of the mean to simulate the distribution of low signal values.

For protein copy number estimations, we assumed total protein mass of 230 pg/cell for BMDMs,[78] 25 pg/cell for non-stimulated and 75 pg/cell for stimulated T cells.[28]

### *In vitro* chronic stimulation of human CD8$^+$ T cells

Human naïve CD8$^+$ T cells were obtained from buffy coats from healthy donors. Total CD8$^+$ T cells were enriched from peripheral blood mononuclear cells by using CD8 MicroBeads (Miltenyi) following manufacturer's instructions. Then, enriched CD8$^+$ T cells were stained with anti-CD45RA Qdot 655 (1/1000, Invitrogen) and anti-CCR7 BV421 (1/80, Biolegend) and naïve CCR7$^+$CD45RA$^+$ T cells were sorted using a FACSAriaIII instrument (BD). Sorted naïve CD8$^+$ T cells were re-suspended in complete RPMI medium (RPMI-1640, 10% FBS, 1X GlutaMax, 1X Peninicilin/Streptomycin, 1X Hepes, 1X non-essential aminoacids, 1X Sodium Pyruvate, 0.01% β-mercaptoethanol) containing rIL-2 (40 U/mL, made in house) and 150,000 cells per well were stimulated with plate-bound anti-CD3 (1 $\mu$g/mL, clone TR66, made in house) and anti-CD28 antibodies (0.5 ug/mL, BD) in 96-well Nunc Cell Culture Plates (ThermoFisher). For chronic stimulation, cells were cultured for 14 days in a humified CO$_2$ incubator at 37 C with and cells were split 1:2 every third day and transferred into plates freshly coated with anti-CD3 and anti-CD28 antibodies. For transient activation naïve CD8$^+$ T cells were stimulated for 3 days and then transferred into 96 U bottom plates and rested in the presence of rhIL-2 only for the remaining 11 days of culture. For re-stimulation experiments, T cells were activated, rested 8 days, and then re-stimulated from day 11 to 14. At day 14 T cells were collected and washed. Some cells were stained with Zombie Green viability dye (Biolegend), anti-PD1 BV785 (1/80, Biolegend), anti-CD39 BV421 (1/100, Biolegend), anti-TIM3 PE-Vio770 (1/20, Miltenyi) and anti-LAG3 PE (1/50, Miltenyi) and analyzed on a FACS Symphony A5 (BD). The remaining cells were snap frozen in liquid nitrogen.

### RNAseq of chronically stimulated T cells and human CD14$^+$ cells

Frozen cell pellets were re-suspended in lysis buffer from the GenElute$^{TM}$ Mammalian Total RNA Miniprep Kit (SIGMA). mRNA was purified following manufacturer's instructions. Then total RNA was quantified by Nanodrop and sequenced at Novogene (Cambridge, United Kingdom) using an Illumina platform (Eukaryotic Transcriptome Library). For initial quality control of sequencing data, FastQC (https://www.bioinformatics.ba) and MultiQC[2] were used. The reads were trimmed to remove sequencing adapters using Trimmomatic[3]. Transcript abundance estimates for all transcripts were performed using Kallisto[4] to generate counts files. Afterwards the count files were imported to R using the 'tximport' R package[5] and summarized to gene level abundances. Only protein-coding genes with valid chromosomal mappings were retained.

### Isolation of OT-I T cells

OT-I T cells were isolated from spleens of Rag$^{-/-}$ HZ OT-I mice. Spleens were disrupted mechanically in MACS buffer and then filtered through 40 $\mu$m cell strainers (BD). Then, cells were centrifuged 5 min at 500 g 4°C and splenocytes were re-suspended in MACS buffer at a concentration of 1 x 10$^7$ cells per mL. CD8$^+$ OT-I T-cells were isolated using the EasySep$^{TM}$ Mouse CD8$^+$ T Cell Isolation Kit (StemCell Technologies) following manufacturer's instructions. Purified OT-I T cells (>95% purity) were re-suspended in complete

RPMI medium supplemented with 10 ng/mL rhIL2 (Sino Biological). Then, OT-I cells were activated with plate-bound anti-CD3$\varepsilon$ (1 μg/mL) and anti-CD28 (0.5 μg/mL) antibodies (Life technologies) 2.2 x10$^6$ cells per plated per well of a 24 well plate.

### CRISPR-Cas9 targeting of genes of interest in T cells

We followed a protocol that was previously established by Lukas Jeker and colleagues.[47,79] FACS-purified T cells (human or mouse) were activated with plate-bound anti-CD3$\varepsilon$ (1 μg/mL) and anti-CD28 (0.5 μg/mL) antibodies. After 24h, T cells were electroporated with a pSpCas9(BB)-2A-GFP plasmid (PX458, Addgene #48138) in which gRNAs targeting different genes were inserted by Golden Gate cloning (see Table S7 for gRNA sequences). Using the Neon Transfection System (Invitrogen), 15 μg of plasmid were electroporated into 2.2 x 10$^6$ 24-h activated T cells in 100 μL transfection tips using the following settings: 1550 V, 3 pulses, 10 ms for mouse OT1-I T-cells and 2200 V, 1 pulse, 20 ms for human CD8$^+$ T cells. Control cells were electroporated with a PX458 vector containing a non-targeting gRNA sequence. After electroporation, T cells were resuspended in complete RPMI supplemented with rhIL-2 (500 U/ml) and plated in 24-well plates coated with anti-CD3$\varepsilon$ (0.5 μg/mL) and anti-CD28 (0.25 μg/mL). Then, GFP$^+$ T cells were sorted and expanded in complete RPMI medium with rhIL-2 (40 U/mL for human, 10 ng/mL for mouse T-cells) for 7-10 days. Gene editing efficiency was analyzed by Tracking of Indels by Decomposition (TIDE) analysis[35]. Briefly, genomic DNA was extracted from control and edited T cells using QuickExtract (Lucigen) and the target loci were amplified by PCR (see list of primers of genomic PCR in Table S7). Amplicons were purified using the NucleoSpin Extract II kit (Macherey Nagel) and sent to Microsynth AG (Switzerland) for Sanger sequencing. Indels and editing efficiency were estimated using the TIDE algorithm by comparing the chromatograms of control and edited sequences. For *in vitro* proliferation assays, expanded T cells were stained with CellTrace Violet (CTV) according to manufacturer's instructions and 100,000 T cells per well (96 well plates) were re-stimulated with plate-bound anti-CD3/anti-CD28 antibodies in the presence of rhIL-2. Cells were analyzed for CTV dilution at different timepoints by flow cytometry (FACS Symphony A5, BD).

### Gene targeting in bone marrow-derived macrophages

Bone marrow progenitor cells were obtained from the tibiae and femurs of ROSA-Cas9 mice and red blood cells were lysed with ACK Lysing Buffer (Life Technologies). Next, bone marrow cells were incubated at 37°C 5% CO$_2$ for 7 days in complete DMEM medium (10% FBS, 1X Penicilin/Streptomycin, 1X GlutaMax) supplemented with 20% of L929 conditioned media containing mM-CSF to induce differentiation toward macrophages (6 x 10$^4$ bone marrow cells per well in 48-well plates, 1 x 10$^6$ cells per well in 6-well plates, 5 x10$^6$ cells per 10 cm petri dish). To edit genes of interest we used the lentiviral vector pKLV-U6gRNA(BbsI)-PGKpuro-2A-BFP (Addgene #50946). Using Golden Gate cloning gRNAs were inserted into the vector (see Table S7 for gRNA sequences). Then, lentiviral particles were generated by transfecting this plasmid into HEK293 cells together with the lentiviral packaging vectors psPAX2 (Addgene #12260) and pMD2.G (Addgene #12259). Lentiviral particles were added to bone marrow progenitors two days after initiating their differentiation towards BMDMs. Five days later (Day 7), medium was removed and adherent BMDMs were cultured in complete medium containing 6 μg/mL of puromycin (LabForce AG) to select transduced BMDMs. The complete medium contained 20% L929 conditioned medium. Four days later, BMDMs were detached for *in vivo* experiments by washing cells with ice cold PBS followed by incubation in PBS-EDTA (5 mM) at 4°C for 40 minutes.

For quantification of IL-12 in supernatants, BMDMs were stimulated with 100 ng/mL LPS (InvivoGen) plus 50 ng/mL rmIFN$\gamma$ (Sino Biological) for 16 h and supernatants were collected and frozen at -80°C. To stain intracellular cytokines in BMDMs, cells were stimulated for 16 h with 100 ng/mL LPS (InvivoGen) and 50 ng/mL rmIFN$\gamma$ (Sino Biological) and then cultured additional 5 h with fresh 100 ng/mL LPS and 1X Brefeldin A (ThermoFisher). Then, cells were detached and stained with anti-F4/80 eFluor 506 (1/100, ThermoFisher), anti-CD86 Super Bright 702 (1/100, ThermoFisher) and anti-MHCII Super Bright 780 (1/500, ThermoFisher) for 30 min at 4°C. Fixation and permeabilization was performed with the True-Nuclear$^{TM}$ Transcription Factor Buffer set (BioLegend) and cells were stained for intracellular cytokines using anti-TNF APC (1/100, ThermoFisher), anti-IL6 eFluor 450 (1/100, ThermoFisher) and anti-IL12 p40 PE (1/100, ThermoFisher). All cells were analyzed on a FACS Symphony A5 (BD). IL-12 in cell supernatants was measured using the Mouse IL-12 p70 DuoSet ELISA kit (R&D Systems).

To assess knockout efficiency, genomic DNA was extracted from 10,000 cells using the QuickExtract reagent (Lucigen). Genomic loci were amplified as described in the T cell section using primers listed in Table S7. Indels and KO efficiency were estimated using the TIDE algorithm[35].

### *In vitro* generation of M1 and M2-polarized human macrophages

CD14$^+$ monocytes were enriched from PBMCs using CD14 MicroBeads (Miltenyi) according to manufacturer's instructions. Monocytes were differentiated and polarized into M1 and M2 macrophages following a published protocol[80]. Briefly, 1 x 10$^6$ monocytes/well were plated in 6-well plates in complete RPMI medium containing either 50 ng/mL hM-CSF or 50 ng/mL hGM-CSF (Sino Biological). After 6 days of differentiation, M-CSF-stimulated macrophages were polarized to M2 by adding fresh medium containing 50 ng/mL rhIL-4 and 50 ng/mL rhIL-13 (Sino Biological). GM-CSF-stimulated macrophages were polarized to M1 by adding 20 ng/mL rhIFN$\gamma$ (Sino Biological) and 50 ng/mL LPS (TLRgradeTM, EnzoLife Sciences). Two days later macrophages were washed with cold PBS and detached by incubating with PBS-EDTA (5mM) at 4°C for 45 minutes. Polarization was confirmed by flow cytometry by staining with antibodies to detect M1 markers: anti-HLA-DR PE-Cy5 (1/50, Beckman Coulter), anti-CD80 APC (1/100, ThermoFisher) and anti-CD86 FITC (1/100, Miltnyi); and M2 markers: anti-CD163 APC (1/100, ThermoFisher) and anti-CD206

 OPEN ACCESS

**Cell Genomics**

PE-Cy7 (1/300, ThermoFisher). Macrophages were washed three times with PBS and pellets were snap frozen in liquid nitrogen and kept at -80°C for later analysis.

### Co-culture of HCC tumor tissue and CD14⁺ monocytes

Tumor and non-tumorous liver tissue from HCC patients was manually cut into small tissue fragments of 1-2 mm³. After processing, tissue fragments were placed in ultra-low adherence Nunclon™ Sphera™ 96-Well U-Shaped-Bottom Microplates in 100 μL of complete medium (RPMI-1640, 10% FBS, 1X GlutaMax, 1X Penicillin/Streptomycin, 1X Hepes, 1X non-essential amino acids, 1X Sodium Pyruvate, 0.01% β-mercaptoethanol). CD14⁺ cells were isolated from peripheral blood from the same patients. CD14⁺ cells were stained using the CellTrace Far Red (CTFR) Cell Proliferation kit (Biolegend) according to the manufacturer's protocol and 50,000 cells CTFR⁺ CD14⁺ cells were added to the wells containing tissue fragments. CTFR⁺ CD14⁺ cells were cultured alone as controls. After 3-7 days of co-culture cells and tissues were recovered, disaggregated mechanically, and filtered through 40 μm cell strainers (BD). Cell suspensions were stained with Zombie UV viability dye (Biolegend), anti-THY1 PE (1/50, Thermo Fisher), anti-CD45 PerCP-Cy5.5 (1/100, Miltenyi), anti-CD11b APC-Vio770 (1/100, Miltenyi) and anti-CD14 V500 (1/50, BD). Stained cells were analyzed using a FACS Symphony A5 instrument (BD).

### Mouse tumor models

B16.OVA cells were provided by Dr. Matteo Bellone (San Raffaele Scientific Institute, Milan) and were maintained in RPMI (Gibco) supplemented with 10% FBS (Gibco), 1X GlutaMax (Gibco), 25 mmol/L HEPES (Gibco) and Penicillin/Streptomycin. MC38 and MC38.OVA murine colon adenocarcinoma cells were provided by Dr. Walter Reith (University of Geneva) and were cultured in DMEM supplemented with 10% heat-inactivated FCS, GlutaMax and 25 mmol/L Hepes. Before injection into mice, cells were trypsinized and washed twice with PBS. Then, 5 x 10⁵ cells were subcutaneously (s.c.) injected in the dorsal region of mice. The size of tumors was measured in a blinded fashion using calipers. For OT-I adoptive transfer experiments, expanded control and CRISPR/Cas9-edited OT-I T-cells were re-stimulated for 48 h in 24-well plates coated with anti-CD3ε/anti-CD28 in the presence of rhIL-2. Then, cells were collected and washed with PBS prior to adoptive transfer. Tumor-bearing mice received 1 x 10⁶ T cells i.v. through the tail vein 5 days after tumor cell inoculation. For co-injection of BMDMs with MC38 cells, cells were mixed in a 1:1 ratio (1 x 10⁶ cells total) and the mixture was injected s.c.

For the analysis of tumor-infiltrating cells, tumors were excised at day 10-12 post tumor cell inoculation and disaggregated mechanically and enzymatically with 2 mg/mL collagenase IV (Roche) and 50 U/mL DNase I (Roche). Single-cell suspensions were filtered in 70 μm cell strainers (BD) and then stained for flow cytometry analysis.

### Flow cytometry

Single-cell suspensions from B16.OVA mouse tumors were stimulated with 50 ng/mL PMA (SIGMA), 1 μg/mL Ionomycin (SIGMA), Brefeldin A and Monensin (Life Technologies) for 5 hours at 37°C. Then, cells were stained with Zombie UV viability dye (Biolegend) to exclude dead cells and then surface markers were stained with the following antibodies: anti-CD8α Super Bright 702 (1/200, ThermoFisher), anti-CD45.1 Super Bright 436 (1/100, ThermoFisher), anti-CD45.2 PerCP-Cy5.5 (1/100, ThermoFisher), anti-PD1 PE (1/200, ThermoFisher), anti-Tim3 PE-Cy7 (1/200, ThermoFisher) and anti-LAG3 FITC (1/200, Thermofisher). Following surface staining, cells were fixed and permeabilized with True-Nuclear™ Transcription Factor Buffer Set (BioLegend) according to the manufacturer's instructions and intracellular staining was performed using anti-IFNγ FITC (1/100, ThermoFisher) and anti-TNF APC (1/100, ThermoFisher).

For detection of *THY1* mRNA in CD14⁺ cells the PrimeFlow™ RNA Assay Kit (Thermo Fisher) was used following the manufacturer's instructions for 96-well plates. Validated target probes were used for THY1 (type 1 Alexa Fluor™ 647, VA1-12482-PF) and GAPDH (type 6 Alexa Fluor™ 750, VA6-10337-PF, positive control).

## QUANTIFICATION AND STATISTICAL ANALYSIS

The number of animals per experimental group was calculated using using the "Power and Sample Size" software from the Department of Biostatistics from the Vanderbilt University (https://vbiostatps.app.vumc.org/). Animals were randomized into different groups after tumor cell inoculation. Investigators were blinded to allocation during experiments and outcome assessment. Statistical analysis was performed using GraphPad Prism8 software (GraphPad Software, Inc.). Two-tailed t-tests were used to compare treatments with control groups; ANOVA or Mixed-Effect analysis models were used to compare continuous outcomes across multiple experimental groups. Tukey corrections were used to adjust P values for multiple comparisons.

## ADDITIONAL RESOURCES

Processed data tables can be visualized in our web platform: www.immunomics.ch/hcc.

