## [Document S2. Transparent peer review records for Canale et al. · Cell Genomics]

Proteomics of immune cells from liver tumors reveals immunotherapy targets

Author list

Fernando P. Canale , Julia Neumann, Janusz von Renesse, Elisabetta Loggi , Matteo Pecoraro, Ian Vogel, Giada Zoppi, Gaia Antonini, Tobias Wolf, Wenjie Jin, Xiaoqin Zheng, Giuliano La Barba, Emrullah Birgin, Marianne Forkel, Tobias Nilsson, Romina Marone, Henrik Mueller, Nadege Pelletier, Lukas T. Jeker, Gianluca Civenni, Christoph Schlapbach, Carlo V. Catapano, Lena Seifert, Adrian M. Seifert, Silke Gillessen, Sara de De Dosso, Alessandra Cristaudi, Nuh N. Rahbari, Giorgio Ercolani, and Roger Geiger

Summary

Initial submission: Received : September 28th 2022

Scientific editor: Judith Nicholson

First round of review: Number of reviewers: 2
Revision invited : November 18th 2022
Revision received : March 13th 2023

Second round of review: Number of reviewers: 2
Accepted : 2nd May 2023

Data freely available: Yes

Code freely available: Yes

This transparent peer review record is not systematically proofread, type-set, or edited. Special characters, formatting, and equations may fail to render properly. Standard procedural text within the editor's letters has been deleted for the sake of brevity, but all official correspondence specific to the manuscript has been preserved.

Referees' reports, first round of review

Reviewer 1

This proteomics project identified both known and novel markers of tumour macrophages and lymphocytes. The evidence for SGPL1 and AFAP1L2 as important and relevant to HCC cancer biology and survival in mice is convincing.

Comments:

1. HCC was compared to 'adjacent liver tissue". Readers need precision in the description of location of each 'adjacent liver tissue" and histopathological images of this tissue. It is likely that non-tumour liver tissue distance from the tumour is relevant to the proteomic profile. Were representative sections of each sample of 'adjacent liver tissue" check by a registered pathologist to ensure that no microtumours or tumour edge was included?
2. The 48 HCC studied included 20 HCV, 9 HBV and 19 non-viral. Details of these deidentified patients are needed; were any on anti-viral therapy? Which 'non-viral ' were alcohol, NAFLD or MAFLD, hemochromatosis or other genetic condition?
3. Figures 2B and 3B show data in order of patient ID number, but ordering by HCV, then HBV then non-viral would allow the reader to better appreciate whether or not there are patterns of expression common to each of these 3 subsets of etiology.
4. Page 6 the sentence, "We then stratified...." Needs greater clarity.
5. Fig 5H and fig 6C data are not compelling, and the claim at the top of page 9 made based on Fig 5H should be deleted .
6. The Discussion is disappointing by its preoccupation with speculation on how the findings might be exploited for therapy. Discussions of the approach used in this paper, comparisons with other omics approaches to understand human HCC biology and other literature relevant to HCC is needed.

Reviewer 2

The study by Canale et al investigates the cancer and immune proteomic profiles of hepatocellular carcinoma. They analyzed a cohort of clinical samples, separated samples to HCC cells, T cells, macrophages, and compared to normal adjacent liver samples and blood cells from the same patients. These analyses highlighted several key regulators of inflammatory responses in the liver, which they went on and validated using in-vivo models. In their analyses they focused on two key proteins, sgpl1, which impacts the M1/M2 macrophage phenotype, and Afap1l2, which impacts T cell exhaustion. Overall, this study presents a very large body of work, including the proteomics of tumor/control subpopulations and several models that were used for validation. It also shows the strength of the team in combining immunology and proteomics, and identifying novel regulators of anti-cancer immune responses. However, despite this large body of work, the manuscript lacks a coherent and clear message. It seems like the manuscript is composed of three distinct stories, one about HCC, second about macrophages and the third about T cells. The focus on liver cancer is very interesting, but eventually I could not understand whether their findings have anything to do with the liver. Furthermore, deeper investigation of each one of these sections is missing and would increase the overall interest in the manuscript.

My comments are listed below:

1. The authors indicate that the cancer/liver samples included mainly hepatocytes, but these samples were mixed with stromal components. It is very difficult to assess the data if we don't know what is the % of cancer/hepatocytes. When the authors write "mainly" hepatocytes, we cannot know if it is >80% or around 50%. Such differences would have major impact on data interpretation. The authors can also correct for such differences and obtain more accurate results.
2. Given the large variance in the number of isolated cells from each sample, the authors have to indicate the number of cells sorted in each sample and explain all the normalizations. It is critical to know that the differences that were seen result from proteomic changes within the cells, and not simply a result of the different numbers of cells in each sample. In relation to this comment, the volcano plot in figure 4 is highly skewed, and suggests that there are major differences between the sample groups, in terms of number of identified proteins. A proper normalization needs to be done prior to these tests.
3. All statistical tests have to be performed with FDR correction. Otherwise these cannot be considered significant. In fact it is surprising that some of these tests

did not result in a higher number of significantly changing proteins. For example, in page 4 they indicate only 234 differentially-expressed proteins between liver and HCC cells. This is quite surprising, since usually, thousands of proteins are found to distinguish between cancer and normal. If this result is due to the large FC cutoff ($\log FC > 2$), then this should be corrected. It is more important to obtain significant changes than basing the results on a random FC cutoff.

4. There is hardly any analysis of the liver/HCC comparison. This is very disappointing, as many analyses can be done with these data to highlight the differences among tumor groups, stages, the association with prognosis or other clinical features, compare to RNA data etc. This section of the manuscript should be substantially expanded in a revised version.

5. In the analyses of immune cells (macrophages and T cells) it seems like the liver context is completely missing. It is not clear why the authors decided to validate their results in MC38 colorectal cancer model using subcutaneous injection, or used the OVA model for the T cells. Using these models distracts from the main topic of the manuscript, which addresses liver cancer and the liver microenvironment. It is not clear whether the roles of the validated proteins are unique to the liver or in general impact cancer immune responses. This is a major problem with the manuscript design, which needs to be addressed. The authors should clearly address the limitation of the models that they used for validation, and perform additional analyses (experimental or analytical) to assess the relevance to the liver.

6. The high induction of TOX, LMCD1 and AFAP1L2 upon chronic stimulation seems to be overestimated. These proteins are seen mostly upon chronic stimulation, and one cannot compare the data to missing or imputed data. The authors should determine these ratios accurately, by targeted MS, using standards, or other antibody-based techniques. They can also delete the FC numbers from the text.

7. NK cells are among the most important immune cells in the liver. Their discussion should be added to the revised manuscript. It is especially important given that they discuss Afap1l2, which was found in their previous work in NK. Furthermore, according to their panel, it seems like they may have isolated NK cells and may have the data. This is an important part of the work that would also be essential to emphasize the liver context.

8. In the analysis of T cells the authors show higher T cell numbers in viral-induced HCC than other. They should investigate whether the differences result from differences in the cancers, or due to the viral infection itself.

9. The authors indicate that in their previous work they saw Afap1l2 in long-term stimulated NK cells. What is the explanation that they did not see the proteins in T cells in the previous study? These discrepancies should be discussed.
 10. Figure 2B the labels are swapped.
 11. Sorting panel (Supplementary Figure 1) should be explained. What are the L-shaped isolations of memory T cells with CCR7 high and low? They also don't refer to all isolated cells in the manuscript.
 12. Statistical significance should be added to the figures.
 13. P.4 they indicate that they isolate hepatocytes. I assume they mean also HCC cells. Should be rephrased.
-

Authors' response to the first round of review

We thank the Reviewers for their constructive criticism and helpful feedback. In response to their suggestions, we added a figure comparing transcriptomic and proteomic data highlighting the nuances of mass spectrometry-based profiling. We also added proteome data on NK cells, extended our online platform, re-arranged figures and extended the Discussion. Please find below our point-by-point responses:

Reviewer #1:

This proteomics project identified both known and novel markers of tumour macrophages and lymphocytes. The evidence for SGPL1 and AFAP1L2 as important and relevant to HCC cancer biology and survival in mice is convincing. We thank the Reviewer for acknowledging the novelty and importance of our findings.

Comments: 1. HCC was compared to 'adjacent liver tissue'. Readers need precision in the description of location of each 'adjacent liver tissue' and histopathological images of this tissue. It is likely that non-tumour liver tissue distance from the tumour is relevant to the proteomic profile. Were representative sections of each sample of 'adjacent liver tissue' checked by a registered pathologist to ensure that no microtumours or tumour edge was included?

The surgeons cut freshly resected specimens into tumorous and non-tumorous tissue based on macroscopic properties. Because the resected tissue was entirely used for immune cell extraction, we could not take any histopathological images and therefore cannot exclude the possibility of microtumors in samples

designated as non-tumorous, as pointed out by the reviewer. To confirm correct classification of tumorous and non-tumorous tissue, we relied on the analysis of their proteomes, which showed that they had a clearly distinct profile (Figures 2A, C), as described on page 5 in the revised manuscript. In addition, our tumorous samples had canonical features of HCC proteomes based on a comparison to a previous proteome study (Jinag et al., see Figure 2A).

2. The 48 HCC studied included 20 HCV, 9 HBV and 19 non-viral. Details of these deidentified patients are needed; were any on anti-viral therapy? Which 'non-viral' were alcohol, NAFLD or MAFLD, hemochromatosis or other genetic condition?

In Table S1 of the revised version of the manuscript, we now provide additional clinical information including tumor staging, comorbidities, and exposure to anti-viral therapies.

3. Figures 2B and 3B show data in order of patient ID number, but ordering by HCV, then HBV then non-viral would allow the reader to better appreciate whether or not there are patterns of expression common to each of these 3 subsets of etiology.

As suggested by the Reviewer, we changed these figures and now show patients ordered by HCV, HBV and then non-viral. In addition, we extended our web-based platform, which now allows analyzing the proteome profiles of these samples in more detail.

4. Page 6 the sentence, "We then stratified...." Needs greater clarity.

We agree that this sentence is not clear and therefore we removed this sentence.

5. Fig 5H and fig 6C data are not compelling, and the claim at the top of page 9 made based on Fig 5H should be deleted.

We agree with the Reviewer that the effect size of data presented in Fig. 5H showing the expression of CD25 is modest. We therefore removed Fig. 5H from the manuscript and no longer report and discuss these data. Regarding Fig. 6C, we found twice as many Afap112-edited OT-I T cells than control OT-I T cells (~18% vs. ~10%) in murine MC38 tumors. We believe this is an important readout, as it helps explain the improved tumor clearance shown in the subsequent sub figures (Figs. 6F-J).

6. The Discussion is disappointing by its preoccupation with speculation on how the findings might be exploited for therapy. Discussions of the approach used in this paper, comparisons with other omics approaches to understand human HCC biology and other literature relevant to HCC is needed.

As requested by the Reviewer, we extended the Discussion to the mass

spectrometry-based proteomics approach and comparisons with other omics approaches relevant to HCC.

Reviewer #2:

The study by Canale et al investigates the cancer and immune proteomic profiles of hepatocellular carcinoma. They analyzed a cohort of clinical samples, separated samples to HCC cells, T cells, macrophages, and compared to normal adjacent liver samples and blood cells from the same patients. These analyses highlighted several key regulators of inflammatory responses in the liver, which they went on and validated using in-vivo models. In their analyses they focused on two key proteins, sgpl1, which impacts the M1/M2 macrophage phenotype, and Afap1l2, which impacts T cell exhaustion. Overall, this study presents a very large body of work, including the proteomics of tumor/control subpopulations and several models that were used for validation. It also shows the strength of the team in combining immunology and proteomics, and identifying novel regulators of anti-cancer immune responses.

We thank the Reviewer for acknowledging the strength of our work, in which we combined immunology and proteomics to identify novel regulators of anti-cancer immune responses.

However, despite this large body of work, the manuscript lacks a coherent and clear message.

We were hoping that our message was clear: We phenotyped T cells and macrophages in liver cancer and identified two novel regulators, AFAP1L2 and SGPL1, which are potential targets for immunotherapy.

It seems like the manuscript is composed of three distinct stories, one about HCC, second about macrophages and the third about T cells.

We agree with the Reviewer that the manuscript reports different aspects of the immune response in HCC. Our contribution was to generate a resource of T cell and macrophage proteomes and to identify novel immunological regulators. Our analysis of HCC and surrounding liver tissue was intended to confirm correct classification of tumorous and non-tumorous tissue, which was made by surgeons based on macroscopic properties. We found that adjacent liver tissues and tumors had clearly distinct profiles with canonical features based on a comparison to a previous proteome study (Jiang et al.).

The focus on liver cancer is very interesting, but eventually I could not understand whether their findings have anything to do with the liver.

We focused on phenotypic changes of immune cells in liver cancer and discovered

two novel immunological regulators, AFAP1L2 and SGPL1, that we found could be generalized to play a role in the immune response to cancer more broadly.

Furthermore, deeper investigation of each one of these sections is missing and would increase the overall interest in the manuscript.

We appreciate the Reviewer's interest in our findings and suggestion for greater mechanistic insights. Ongoing work in our lab is focused on characterizing the mechanism by which AFAP1L2 modulates T cell function. With regard to this manuscript, we hope that our datasets and findings provide a rich resource to the community. For example, the proteome data on cancer and liver tissue are now accessible for deeper interrogation on our improved online platform.

My comments are listed below: 1. The authors indicate that the cancer/liver samples included mainly hepatocytes, but these samples were mixed with stromal components. It is very difficult to assess the data if we don't know what is the % of cancer/hepatocytes. When the authors write "mainly" hepatocytes, we cannot know if it is >80% or around 50%. Such differences would have major impact on data interpretation. The authors can also correct for such differences and obtain more accurate results.

We agree with the Reviewer that the exact percentage of hepatocytes in our samples is unknown. Therefore, we changed the text to: "we recovered a cell pellet from the Ficoll gradient, which contains hepatocytes, malignant and stromal cells".

2. Given the large variance in the number of isolated cells from each sample, the authors have to indicate the number of cells sorted in each sample and explain all the normalizations. It is critical to know that the differences that were seen result from proteomic changes within the cells, and not simply a result of the different numbers of cells in each sample. In relation to this comment, the volcano plot in figure 4 is highly skewed, and suggests that there are major differences between the sample groups, in terms of number of identified proteins. A proper normalization needs to be done prior to these tests.

In the revised version of the manuscript, we now provide the number of sorted cells per sample in Supplementary Table 1. Regarding the direct comparison of proteomes, the number of identified proteins depends on the number of cells as pointed out by the reviewer, but also on their phenotype, sample preparation and instrument performance. We report the number of identified proteins per sample in Figure 2D. For the vast majority of samples, we quantified more than 4,000 proteins. The volcano plot shown in Figure 4 is skewed towards PD-1 expressing

cells, because as proliferating cells their mass is bigger than that of resting cells. The skewed volcano plot thus reflects the underlying biology. Regarding normalizations, we used normalized values generated by MaxQuant referred to as “Label Free Quantitation” throughout. By additional median normalization, the volcano plot would become more symmetric but, in our opinion, this would be an over-normalization.

3. All statistical tests have to be performed with FDR correction. Otherwise these cannot be considered significant. In fact it is surprising that some of these tests did not result in a higher number of significantly changing proteins. For example, in page 4 they indicate only 234 differentially-expressed proteins between liver and HCC cells. This is quite surprising, since usually, thousands of proteins are found to distinguish between cancer and normal. If this result is due to the large FC cutoff ($\log FC > 2$), then this should be corrected. It is more important to obtain significant changes than basing the results on a random FC cutoff.

Regarding the HCC tissue data, our prior cutoffs resulted in 234 differently abundant proteins. To avoid confusion, we no longer report the number of differently abundant proteins in the revised version of the manuscript. As pointed out by the Reviewer, FDR correction of P-values along with other methods is useful for controlling the rate of false positives when conducting multiple comparisons. However, during the revision of the manuscript we compared proteomes of liver and tumor NK cells, which revealed differently abundant proteins with relatively low P-values due to variation between patient samples. Consequently, upon FDR adjustment no proteins met a strict threshold of an FDR-adjusted P-value below 0.05. Importantly, we found that AFAP1L2 is one of the most upregulated proteins in tumor-infiltrating NK cells, which we consider relevant (Figures 4A, B). Hence, we show P-values in this figure and for consistency maintain P-values from t-tests (with Welch correction) without FDR adjustments in all other figures. To allow readers performing their own analysis of our datasets we introduced a feature in our online platform that allows for adjusting P-values according to different methods (FDR, Bonferroni, Hochberg) (Figure 1 for the Reviewer).

Figure 1: Comparison of proteomes of liver and tumor macrophages with FDR correction of P values. The values for GLUL are shown. FC: -4.63 ; -Log10 P-value 4.6.

4. There is hardly any analysis of the liver/HCC comparison. This is very disappointing, as many analyses can be done with these data to highlight the differences among tumor groups, stages, the association with prognosis or other clinical features, compare to RNA data etc. This section of the manuscript should be substantially expanded in a revised version.

We are pleased that the Reviewer is interested in further analyses of the liver/HCC proteome data. Therefore, we extended our online platform and now include features to compare liver and HCC proteomes of patients with different disease etiologies and stages, which we hope will inspire future research into liver cancer. Since our main focus was to identify and validate potential targets for immunotherapy, we did not further investigate aspects related to HCC biology.

5. In the analyses of immune cells (macrophages and T cells) it seems like the liver context is completely missing. It is not clear why the authors decided to validate their results in MC38 colorectal cancer model using subcutaneous injection, or used the OVA model for the T cells. Using these models distracts from the main topic of the manuscript, which addresses liver cancer and the liver microenvironment. It is not clear whether the roles of the validated proteins are 5 unique to the liver or in general impact cancer immune responses. This is a major problem with the manuscript design, which needs to be addressed. The authors

should clearly address the limitation of the models that they used for validation, and perform additional analyses (experimental or analytical) to assess the relevance to the liver.

This project characterizes immune cell phenotypes via proteomics in a cohort of HCC patients. We discovered novel immunological regulators that we found could be generalized to play a role in the immune response to cancer more broadly. We cross-referenced our data with other resources from different tumor types and found that AFAP1L2 is also induced in T cells in CRC and NSCLC (Suppl Fig. 4). This is not surprising because our in vitro data further supported AFAP1L2 upregulation as a feature of chronic T cell stimulation. Similarly, we found that SGPL1 was induced by canonical M2-polarizing cytokines, IL-4 and IL-13. Thus, we employed well-established tumor mouse models (MC38 and B16) for cancer immunotherapy studies. We demonstrate that ablation of AFAP1L2 increases the activity and survival of repeatedly stimulated T cells and that ablation of SGPL1 enhances the inflammatory phenotype of macrophages. Thus, AFAP1L2 and SGPL1 regulate the activity of immune cells, which affects the antitumor immune response against multiple tumor types.

6. The high induction of TOX, LMCD1 and AFAP1L2 upon chronic stimulation seems to be overestimated. These proteins are seen mostly upon chronic stimulation, and one cannot compare the data to missing or imputed data. The authors should determine these ratios accurately, by targeted MS, using standards, or other antibody-based techniques. They can also delete the FC numbers from the text. In Figure 5C, we show Log₂ LFQ (label-free quantitation) values.

MaxLFQ is an intensity determination and normalization procedure and LFQ values are typically used to compare protein abundances between samples. For more details see Cox et al., MCP 2014. Based on the LFQ intensities, we estimated protein copy numbers of AFAP1L2 using previously described methods (Wolf et al. NI 2020). These data are shown in Figures 5D and E. While transiently activated T cells contain close to zero copies of AFAP1L2, chronically activated T cells contain approximately 300,000 copies. These are comparisons of absolute intensities or absolute copy numbers, and we did not report FC values (fold changes).

7. NK cells are among the most important immune cells in the liver. Their discussion should be added to the revised manuscript. It is especially important given that they discuss Afap1l2, which was found in their previous work in NK. Furthermore, according to their panel, it seems like they may have isolated NK cells and may have the data. This is an important part of the work that would also

be essential to emphasize the liver context.

We agree that NK cells play an important role in the liver. Indeed, we isolated NK cells from 24 HCC patients and added this proteome data to the revised version of the manuscript. Importantly, a comparison of liver and tumor NK cells showed that AFAP1L2 was among the most upregulated proteins in tumor NK cells. These data are shown in Figures 4A, B of the revised manuscript and mentioned in the discussion.

8. In the analysis of T cells the authors show higher T cell numbers in viral-induced HCC than other. They should investigate whether the differences result from differences in the cancers, or due to the viral infection itself.

We observed a higher absolute number of T cells in tumors of viral etiology as well as higher frequencies of PD-1+ T cells. Whether this association is caused by the viral infection itself or due to intrinsic properties of the tumors is an interesting question, which would be difficult to resolve from our patient cohort. On our revised web platform, we added a feature to investigate differences in protein abundance between viral and non-viral HCCs. A detailed investigation into this is beyond the scope of this manuscript.

9. The authors indicate that in their previous work they saw Afap1l2 in long-term stimulated NK cells. What is the explanation that they did not see the proteins in T cells in the previous study? These discrepancies should be discussed.

We show in Figures 5D-F that AFAP1L2 is not present in T cells in response to canonical activation conditions, which we also used in our previous studies (48 h exposure to plate-bound CD3 and CD28 antibodies). In contrast, only when T cells are re-stimulated or chronically stimulated (exposure to CD3 and CD28 antibodies for 14 days) AFAP1L2 is upregulated. These results are fully consistent with our previous observations.

10. Figure 2B the labels are swapped.

Thank you for pointing out this mistake. This is corrected in the revised version of the manuscript.

11. Sorting panel (Supplementary Figure 1) should be explained. What are the L-shaped isolations of memory T cells with CCR7 high and low? They also don't refer to all isolated cells in the manuscript.

The L-shaped gate includes all memory T cell subsets and excludes naïve T cells. For clarifications, we changed the text in the captions of Figure S1 to: "Shown are flow cytometry plots and gating strategy to sort CD14+ monocytes/macrophages, total memory CD4+ and CD8+ T cells and CD56+ NK cells. The L-shaped gate on T

cells includes CCR7+CD45RA- central memory T cells, CCR7- CD45RA- effector memory T cells and CCR7- CD45RA- EMRA T cells (Sallusto et al., 2004).” In addition, we added a gate on NK cells.

12. Statistical significance should be added to the figures.

We added statistical significance to the Figures 4B, 5E, 5F and S4C.

13. P.4 they indicate that they isolate hepatocytes. I assume they mean also HCC cells. Should be rephrased.

We rephrased the sentence on page 4 to, “we recovered a cell pellet from the Ficoll gradient, which contains hepatocytes, malignant and stromal cells (Figure 1A).”

Referees' report, second round of review

Reviewer 1

Recommends accept

Reviewer 2

I was disappointed to see that most of my major comments were not properly addressed in the revised manuscript. The only major comment that was addressed was the addition of NK cells data, which is appreciated by this reviewer. However, the structure of the manuscript remained unchanged, and includes multiple analyses in different directions that do not integrate into a coherently flowing story. Specifically, the manuscript starts with an interesting liver cancer cohort, but moves away from that to unrelated systems; then the analyses of T-cells, NK-cells and macrophages are not connected mechanistically to explain the focus on the specific proteins. I do appreciate the value of the resource, but the authors should have made an effort to provide a broader system's view instead of jumping directly to multiple small stories.

Beyond the structural comments, the authors also did not address the analytical and statistical comments. Specifically, the cancer/stroma mix is a major drawback that had to be addressed, not only omit a sentence from the text. Having different ratios of stroma and liver/cancer cells might mask true differences between samples, potentially leading to low statistical significance of liver-cancer comparisons. The authors should have tried to deconvolute the data to at least estimate the contribution of stroma to each sample, and then correct for this

difference. In terms of the statistical analyses, the statistical tests and cutoffs should be indicated for each analysis. I do not accept the explanation that the authors avoided FDR control because of the NK cells. If one of their analyses is insignificant, they should not compromise the statistics of all other tests. The addition of a web tool to analyze the data is valuable, but it cannot come instead of actually performing the analyses themselves.

I think the manuscript will benefit from substantial restructuring and correction of some of the computational analyses as indicated above and in my previous comments.

Authors' response to the second round of review

Reviewer #2:

I was disappointed to see that most of my major comments were not properly addressed in the revised manuscript. The only major comment that was addressed was the addition of NK cells data, which is appreciated by this reviewer. However, the structure of the manuscript remained unchanged, and includes multiple analyses in different directions that do not integrate into a coherently flowing story. Specifically, the manuscript starts with an interesting liver cancer cohort, but moves away from that to unrelated systems; then the analyses of T-cells, NK-cells and macrophages are not connected mechanistically to explain the focus on the specific proteins. I do appreciate the value of the resource, but the authors should have made an effort to provide a broader system's view instead of jumping directly to multiple small stories.

As discussed over Zoom (with the editor), we look forward to receiving your edits and suggestions for further improvement of the manuscript. *editors note – discussed redrafting the manuscript with the authors for improved clarity and flow.

Beyond the structural comments, the authors also did not address the analytical and statistical comments. Specifically, the cancer/stroma mix is a major drawback that had to be addressed, not only omit a sentence from the text. Having different ratios of stroma and liver/cancer cells might mask true differences between samples, potentially leading to low statistical significance of liver-cancer

comparisons. The authors should have tried to deconvolute the data to at least estimate the contribution of stroma to each sample, and then correct for this difference.

In the revised version of the manuscript, we included a paragraph in the Discussion that highlights the limitations of our study, specifically the inability to determine the cancer cell/stroma ratio in tumor samples.

In terms of the statistical analyses, the statistical tests and cutoffs should be indicated for each analysis. I do not accept the explanation that the authors avoided FDR control because of the NK cells. If one of their analyses is insignificant, they should not compromise the statistics of all other tests. The addition of a web tool to analyze the data is valuable, but it cannot come instead of actually performing the analyses themselves.

As discussed, we added a new Supplementary Figure 2B to the revised version of the manuscript, which explains how to navigate the online platform, including how to make p-value adjustments.

I think the manuscript will benefit from substantial restructuring and correction of some of the computational analyses as indicated above and in my previous comments.